# Precise mapping of single-stranded DNA breaks by sequence-templated erroneous DNA polymerase end-labelling

Leonie Wenson [1], Johan Heldin [1], Marcel Martin [2], Yücel Erbilgin[3], Barış Salman [3,4], Anders Sundqvist[1], Wesley Schaal [1], Friederike A. Sandbaumhüter[1], Erik T. Jansson[1], Xingqi Chen [5], Anton Davidsson[5], Bo Stenerlöw [5], Jaime A. Espinoza [1], Mikael Lindström [6], Johan Lennartsson[1], Ola Spjuth [1] & Ola Söderberg [1] ✉

The ability to analyze whether DNA contains lesions is essential in identifying mutagenic substances. Currently, the detection of single-stranded DNA breaks (SSBs) lacks precision. To address this limitation, we develop a method for sequence-templated erroneous end-labelling sequencing (STEEL-seq), which enables the mapping of SSBs. The method requires a highly error-prone DNA polymerase, so we engineer a chimeric DNA polymerase, Sloppymerase, capable of replicating DNA in the absence of one nucleotide. Following the omission of a specific nucleotide (e.g., dATP) from the reaction mixture, Sloppymerase introduces mismatches directly downstream of SSBs at positions where deoxyadenosine should occur. This mismatch pattern, coupled with the retention of sequence information flanking these sites, ensures that the identified hits are bona fide SSBs. STEEL-seq is compatible with a variety of sequencing technologies, as demonstrated using Sanger, Illumina, PacBio, and Nanopore systems. Using STEEL-seq, we determine the SSB/base pair frequency in the human genome to range between 0.7 and $3.8 \times 10^{-6}$ with an enrichment in active promoter regions.

Cellular DNA is constantly subjected to threats, both from the environment via exposure to radiation or chemicals and as a consequence of natural cellular processes, e.g., the production of endogenous reactive oxygen species or DNA replication and transcription[1]. In the worst-case scenario, this can cause DNA lesions, which may manifest as chemical modifications of nucleobases or strand breaks. Single-stranded DNA breaks (SSBs) are the most common type of DNA damage, occurring at a frequency of one SSB per cell every 1–10 s[1]. Although this type of damage is not as severe as double-stranded DNA breaks (DSBs), SSBs are

transformed into DSBs when independent SSBs on opposite DNA strands are in close proximity. The repair of SSBs shows high fidelity, as the opposite DNA strand serves as the template for the missing nucleotide. In contrast, the repair of DSBs causes losses in genetic material and may result in chromosomal translocation[2,3]. In the case of damaged nucleobases, specific DNA excision repair pathways remove the damaged nucleotide and convert the lesion into an SSB[4]. Although the repair of SSBs is highly efficient, this process involves a risk of introducing mutations. This has been exploited by B cells, which

[1]Department of Pharmaceutical Biosciences, Science for Life Laboratory, Uppsala University, Biomedical Center, Uppsala, Sweden. [2]Department of Biochemistry and Biophysics, National Bioinformatics Infrastructure Sweden, Science for Life Laboratory, Stockholm University, Solna, Sweden. [3]Aziz Sancar Institute of Experimental Medicine, Department of Genetics, Istanbul University, Istanbul, Türkiye. [4]Institute of Graduate Studies in Health Sciences, Istanbul University, Istanbul, Türkiye. [5]Department of Immunology, Genetics & Pathology, Science for Life Laboratory, Uppsala University, Biomedical Center, Uppsala, Sweden. [6]Division of Genome Biology, Department of Medical Biochemistry and Biophysics, Science for Life Laboratory, Karolinska Institutet, Stockholm, Sweden. ✉e-mail: ola.soderberg@uu.se

introduce somatic hypermutations during the affinity maturation of antibodies. Furthermore, translesion DNA polymerase η fills in the gaps in immunoglobulin genes, generated by the concerted action of the enzymes AID, UDG, and APE1, in an error-prone manner [5].

Methods that efficiently detect DNA damage, such as SSBs, DSBs, or damaged nucleobases, are necessary for accurately determining the consequences of a damaging agent, as well as the subsequent extent of repair. Examples of current methods that provide a gross estimate of DNA fragmentation include the COMET assay[6] and visualization of DNA damage in cells via the incorporation of fluorophore-labeled nucleotides[7]. We have recently developed a method that combines different DNA polymerases to label SSBs or DSBs selectively[8]. Recent years have seen the development of sequencing-based assays that utilize end-labeling[9], either via methylated nucleotides (RADAR-seq)[10] or synthetic noncanonical deoxyribonucleotides (DENT-seq)[11]. Moreover, several of the approaches that exist for the detection of DSBs, including BLESS or BLISS, are based on end-labeling[12]. However, most current methods used to detect SSBs require extensive manipulation of the extracted DNA, including sonication and heat denaturation, which create secondary DNA breaks that do not accurately represent the events that occur in the cellular environment. Methods such as SSiNGLe[13], Nick-seq[14], and GLOE-seq[15] label the 3′-OH end of an SSB by adapter ligation or via terminal deoxynucleotidyl transferase (TdT). However, as free 3′-OH ends are present in both SSBs and DSBs, these approaches cannot exclusively identify SSBs[16]. Furthermore, no methodology can precisely determine whether multiple sequencing reads originate from different cells or represent multiple PCR amplifications of one molecule. Thus, there is currently a pressing need for methods that provide precise descriptions of SSBs, i.e., with a precision of a few nucleotides, to identify where these lesions are in the genome and whether the identified breaks are shared between cells.

To address this need, we developed a robust method for detecting SSBs. We envisioned using a highly error-prone DNA polymerase to initiate DNA synthesis from an SSB when only three types of nucleotides are present, for example, when dATP is omitted from the nucleotide mixture. Hence, this DNA polymerase would introduce mismatches directly downstream from any detected SSBs (Fig. 1). The inclusion of biotinylated nucleotides would enable researchers to extract only modified DNA regions. The DNA could subsequently be cut using any restriction enzyme, which would cut the DNA outside of the modified regions due to mismatches, thereby enabling the ligation of sequencing adapters. This method would facilitate sequence-

templated erroneous end-labelling sequencing (STEEL-seq) of SSBs, conferring a unique stretch of modified nucleotides on each individual molecule. The sequence reads will contain sequence information downstream from the SSBs that will validate the presence of bona fide SSBs. The DNA polymerase required for this STEEL-seq method, i.e., the detection of SSBs, needs several features. Namely, it must be highly error-prone, have the ability to extend from a mismatched base, tolerate modified nucleotides (e.g., those that are biotinylated), lack 3′−5′ exonuclease activity (i.e., no proof-reading), yet demonstrate 5′−3′ exonuclease activity to remove the strand in front of the nick while synthesizing a new strand.

DNA polymerases in eukaryotes and prokaryotes can be divided into seven different families, based on sequence homology: A, B, C, D, X, Y, and RT[17]. Numerous specialized DNA polymerases are responsible for DNA replication and maintenance[18]. Despite these specialized roles, most DNA polymerases share a common structure and mechanism of action. For instance, these enzymes fold into a conformation that resembles the shape of a right hand, comprising fingers, palm, and thumb subdomains[19,20]. Binding to DNA via the thumb subdomain elicits a structural change, upon which the thumb and fingers form a complete circle around the DNA strand. The palm subdomain of the polymerase contains the active site, which interacts with the template strand along the minor groove. In contrast, the fingers subdomain is essential for nucleotide recognition and binding[19]. DNA polymerases also share other common traits, such as the ability to synthesize DNA in the 5′−3′ direction only. Moreover, they require all four dNTPs, a DNA template, and a DNA primer from which to initiate synthesis, as well as divalent magnesium ($Mg^{2+}$) as a cofactor[21]. There are, however, some exceptions to these requirements, such as TdT, which does not require a template for DNA synthesis[22]. This special feature makes TdT a valuable tool in many biotechnological applications. Polymerases belonging to the A, B, C, D, and RT families are, with some exceptions, primarily specialized in replication and repair[20]. Replicative polymerases are highly selective during dNTP incorporation and even demonstrate 3′−5′ exonuclease activity (proofreading), which checks the newly synthesized strand and excises any mismatches that may have occurred during replication. The high fidelity of DNA synthesis, when considered together with proofreading functions, means that most polymerases demonstrate very low error rates[23]. A consequence of this elevated selectivity is a decreased tolerance to DNA lesions, which prevent replicative polymerases from bypassing a lesion, resulting instead in replication fork stalling and collapse[23,24]. Other

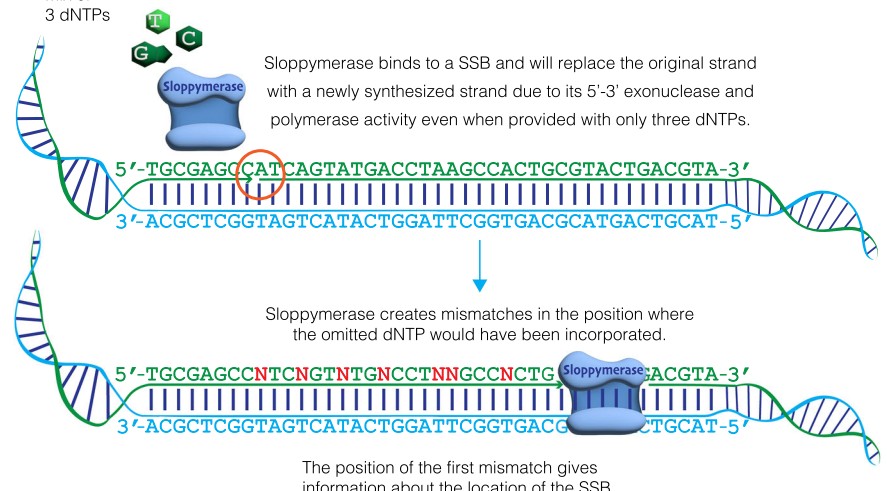

Mix of 3 dNTPs

Sloppymerase binds to a SSB and will replace the original strand with a newly synthesized strand due to its 5'-3' exonuclease and polymerase activity even when provided with only three dNTPs.

5′-TGCGAGCCATCAGTATGACCTAAGCCACTGCGTACTGACGTA-3′
3′-ACGCTCGGTAGTCATACTGGATTCGGTGACGCATGACTGCAT-5′

Sloppymerase creates mismatches in the position where the omitted dNTP would have been incorporated.

5′-TGCGAGCNTCNGTNTGNCCTNNGCCNCTG...ACGTA-3′
3′-ACGCTCGGTAGTCATACTGGATTCGGTGACG...CTGCAT-5′

The position of the first mismatch gives information about the location of the SSB.

**Fig. 1 | Overview of the STEEL-seq method.** Sloppymerase binds to a single-strand break (SSB) on a DNA strand. The strand downstream of the nick is degraded in the 5′−3′ direction by the exonuclease activity of Sloppymerase, while the polymerase activity enables the incorporation of three dNTPs, with dATP omitted from the dNTP mixture. This creates mismatches (N) in positions opposite of dT.

polymerases possess a more open active site, which enables the replication of damaged DNA templates without stalling. This process, referred to as translesion synthesis, is facilitated by polymerases belonging to the Y family (which includes DNA polymerase η)[20,25]. In the case of Y-family polymerases, the DNA-binding surface is extended by a polymerase-associated domain[26], a little finger domain. At the same time, the selection of incoming dNTPs is guided by hydrogen-bond complementarity rather than shape[27]. X-family polymerases are specialized in small gap-filling repair synthesis during base excision repair and non-homologous end joining (NHEJ)[28].

Several mutated DNA polymerases with reduced fidelity have been reported[29,30]; however, we opted to develop an engineered chimeric DNA polymerase with even lower fidelity. We describe herein how we designed the DNA polymerase—Sloppymerase—and assessed its performance when one or more nucleotides were omitted from the dNTP mixture. We then utilized Sloppymerase to develop the STEEL-seq method and tested its performance using various sequencing technologies.

## Results

### Design of Sloppymerase

To ensure the robust detection of SSBs, the DNA polymerase applied to the STEEL-seq method must have several features. Namely, highly error-prone functioning, the ability to extend from a mismatched base, high tolerance to modified nucleotides (e.g., biotinylated nucleotides), a lack of 3′–5′ exonuclease activity (i.e., no proof-reading), and 5′–3′ exonuclease activity to remove the strand in front of the nick while synthesizing a new strand. This combination of features has not yet been found in any DNA polymerase that has evolved in the natural world; this is unsurprising, as such a DNA polymerase would not provide the faithful replication of organismal DNA. Hence, we set out to engineer a DNA polymerase that would demonstrate all of the aforementioned specifications.

DNA polymerases have evolved to exhibit high fidelity, thereby minimizing the error rate during replication. The disadvantage of this high fidelity is that DNA polymerases are unable to bypass damaged bases, e.g., cis-syn thymine dimers, abasic sites, and 7,8-dihydro-8-oxo-guanine (8-oxoG), when replicating DNA. To overcome this limitation, both prokaryotes and eukaryotes have specific translesion DNA polymerases that can replicate DNA, templated by such damaged bases. As a consequence of the relaxed requirement for nucleotide identification, translesion DNA polymerases exhibit higher frequencies of base misincorporation when applied to undamaged DNA. For instance, the translesion DNA polymerase η has a base substitution error rate of approximately $10^{-2}$ to $10^{-3}$. However, polymerase η demonstrates low processivity, incorporating only a few nucleotides before falling off the DNA strand; moreover, this polymerase does not have 5′–3′ exonuclease activity[31]. Hence, to be applicable to our research, DNA polymerase η would need domains that possess 5′–3′ exonuclease activity; *Escherichia coli* DNA polymerase I demonstrates this type of activity, which is necessary for removing the RNA primer of Okazaki fragments during replication and filling gaps in the lagging strand. This DNA polymerase I contains three domains: the first has 5′–3′ exonuclease activity for the removal of the RNA or DNA strand ahead of polymerization, the second exhibits 3′–5′ exonuclease activity for proofreading, and the third is responsible for DNA polymerase activity. DNA polymerase η from *S. cerevisiae* was codon-optimized for expression in *E. coli* and fused, via a 4 × TGS spacer, to the 5′–3′ exonuclease domain (nucleotides 1–969) of *E. coli* DNA polymerase I. This yielded our engineered chimeric DNA polymerase, called Sloppymerase, with the full sequence detailed in Supplementary Fig. 1. Sloppymerase was equipped with a His-tag at the 5′ end for subsequent purification and was expressed in *E. coli* under the control of an L-arabinose-inducible promoter. The purified Sloppymerase had a predicted molecular weight (Mw) of 109 kDa, and its identity was validated by Western blot and mass spectrometry (Supplementary Figs. 2 and 3).

### Characterization of Sloppymerase

To test the performance of Sloppymerase, we utilized a nicked DNA hairpin, as it provides a straightforward approach to determine both 5′–3′ exonuclease and polymerase activity via denaturing polyacrylamide gel electrophoresis (PAGE). A nicked DNA hairpin, when subjected to denaturing PAGE, will separate into two fragments: one small fragment consisting of the 3′ end of the nicked hairpin and one large fragment consisting of the 5′ end of the nicked hairpin. 5′–3′ exonuclease activity can then be monitored based on the degradation of the small fragment, while polymerase activity can be monitored based on the increasing size of the large fragment. We first determined whether Sloppymerase can extend the hairpin by using all four dNTPs, after which we omitted dCTP from the dNTP mixture. The data confirm that Sloppymerase possesses both 5′–3′ exonuclease and polymerase activities, and is also able to polymerize DNA when dCTP is missing from the nucleotide mixture (Fig. 2A). The omission of dCTP, however, noticeably decreases Sloppymerase speed. When all four dNTPs were present in the mix, a full-length band appeared after 5 min, indicating the incorporation of 54 nucleotides, which corresponds to a speed of 10 nucleotides per minute. Next, we assayed how the removal of each of the four dNTPs affects the polymerase activity of Sloppymerase. The omission of dATP only slightly reduced the efficiency of Sloppymerase relative to what was observed when all four dNTPs were present. The removal of dCTP from the dNTP mixture further slowed down the speed of Sloppymerase, and the omission of either dTTP or dGTP resulted in the slowest Sloppymerase speed. Interestingly, Sloppymerase even works when only two types of nucleotides (dGTP and dTTP) are available, although the efficiency drops noticeably (Fig. 2B). The data clearly show that Sloppymerase can replicate DNA when one or more of the four required nucleotides are omitted. An illustration of the predicted folding of Sloppymerase, generated with AlphaFold[32,33], is presented in Fig. 2C.

Having demonstrated that Sloppymerase can modify the sequence downstream of SSBs, we next investigated whether the engineered polymerase exhibits any bias in base substitutions. We therefore ligated an adapter to the extended hairpin and performed PCR using primers specific for the loop of the hairpin and the adapter. The PCR product was cloned into TOPO vectors and transformed into *E. coli* bacteria. Single colonies were collected and sequenced using Sanger sequencing. For a graphic presentation of the experimental design, see Supplementary Fig. 4. The extended hairpins synthesized with all four dNTPs present exhibited few alterations in the predicted sequence (Fig. 3A, top). In contrast, the extended hairpins synthesized when dATP was omitted from the dNTP mixture (Fig. 3A, left) shared the characteristic that deoxyadenosine was mainly replaced with deoxyguanosine. In addition, several insertions and deletions were observed. We also performed Sanger sequencing of extended hairpins that had been synthesized when dCTP was omitted from the dNTP mixture (Fig. 3A, right). In this case, deoxycytidine was mainly replaced with deoxythymidine while several insertions and deletions were also observed. Hence, the data show that Sloppymerase introduces other nucleotides when the correct one is missing, with a clear bias to incorporate dTTP when dCTP is missing, along with the incorporation of dGTP when dATP is missing. In other words, purines are replaced with purines, while pyrimidines are replaced with pyrimidines, i.e., transitions rather than transversions. Sloppymerase also introduces deletions and insertions. As a high degree of variation in the substitutions, as well as the length of Sloppymerase-modified extended hairpins, was observed, we needed to determine whether our strategy produces unique sequence modifications downstream of SSBs. Hence, we performed Illumina (MiSeq Nano) sequencing to generate a comprehensive representation of the possible sequences. We designed a hairpin that allowed us to sequence 79 nucleotides, where the last 55

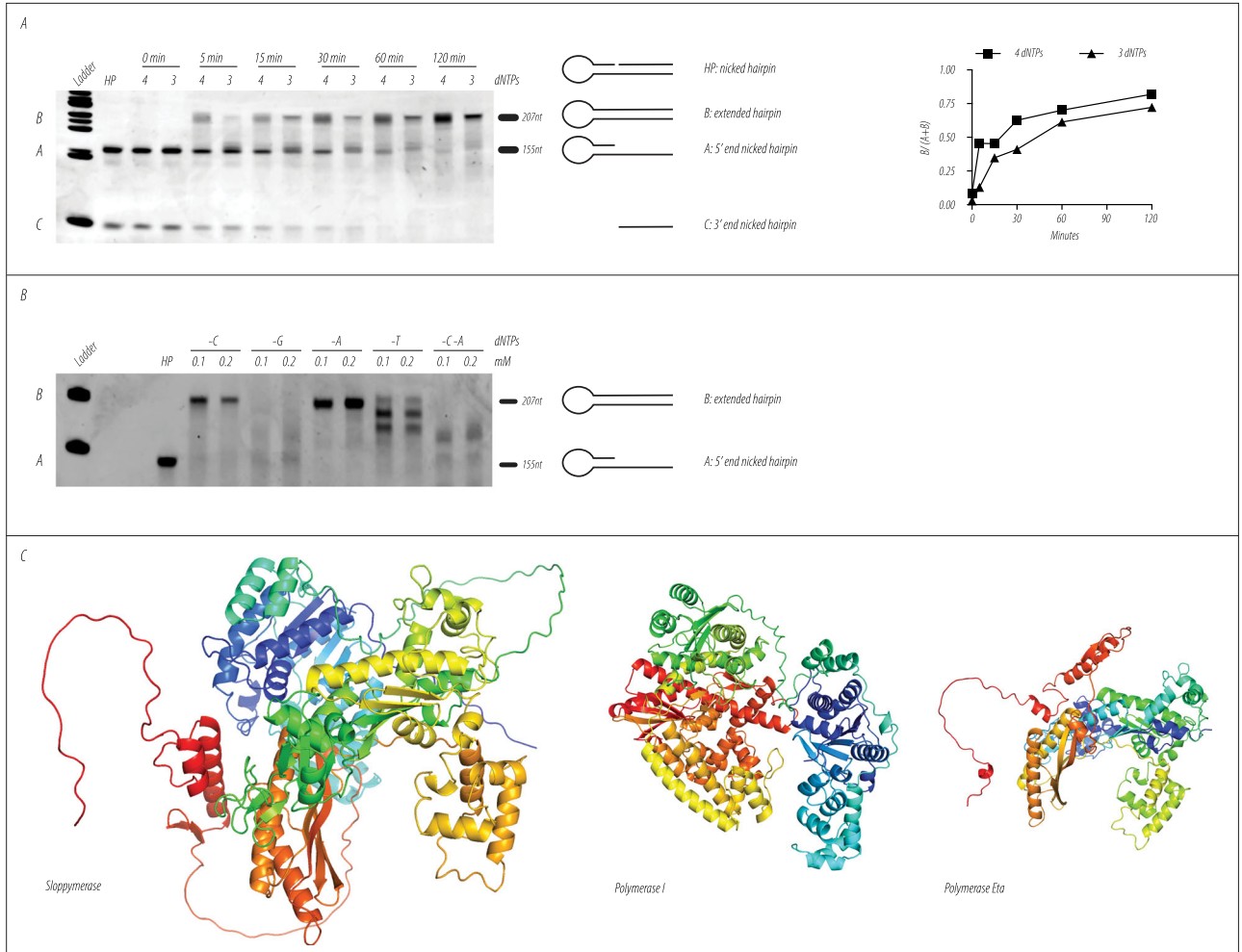

**Fig. 2 | Determination of Sloppymerase characteristics. A** Time lapse experiment to assess Sloppymerase processivity. A nicked hairpin (HP) was incubated with either all dNTPs present (4) or with dATP, dGTP, and dTTP present (3), with DNA synthesis followed at different time points (0–120 min). When subjected to denaturing PAGE, the hairpin dissociates into two fragments: a longer 5′ fragment (**A**) and a shorter 3′ fragment (**C**). Extension of the longer fragment (**B**) indicates DNA polymerase activity, while degradation of fragment C indicates 5′–3′ exonuclease activity. A quantification of the bands (**A** and **B**) in the gel image are presented, which do not take in account the smear in between that indicate partially extended hairpins. The actual efficiency is hence even higher. **B** Processivity of Sloppymerase is dependent on the combination of provided dNTPs. A nicked hairpin (HP) was incubated with Sloppymerase and different combinations (either one or two dNTPs were omitted) and concentrations of dNTPs. Bands in position (**B**) imply full extension of the hairpin, whereas bands in position (**A**) indicate no elongation. **C** Structure of Sloppymerase, *E. coli* DNA polymerase I and DNA polymerase η, as predicted by AlphaFold.

nucleotides had been modified by Sloppymerase (Fig. 3B). In the presence of all four dNTPs, we determined the error rate (+/− SD) to be 4.6+/−3.4%. The error rate was highest for A: 9.0+/−3.3%, followed by T: 4.6+/−3.0%, G: 3.1+/−1.3% and C: 2.4+/−0.8%. When dATP was omitted from the dNTP mix, Sloppymerase introduced one of the other three nucleotides, most frequently G, or jumped over the position. Deletion was the most frequent choice at positions with multiple A's. Of the 124,452 mapped reads, 98,097 were identified as unique sequences (79%), containing different substitutions, insertions, and deletions. Although we designed the experimental setup to favor PCR amplification of the Sloppymerase-modified strand, we observe around 10% of the reads that originate from PCR products of the template DNA strand with no substitutions of A:s. So, the actual percentage of unique sequences is even higher. For the situation when all dNTPs were present, the 448,018 mapped reads gave 232,260 unique sequences (52%).

## Sequence-templated erroneous end-labelling sequencing (STEEL-seq)

Having confirmed that Sloppymerase can modify the DNA downstream of SSBs, we decided to sequence Sloppymerase-treated human DNA using PacBio (Sequel, HiFi reads), Nanopore (PromethION), and Illumina (NovaSeq 6000) sequencing to show that the Sequence-templated erroneous end-labelling sequencing (STEEL-seq) strategy (Fig. 1) is applicable to the major sequencing platforms. DNA collected from TK6 (for Illumina and PacBio) or HaCaT cells (for Nanopore) was treated with the Nickase Nt.BsmAI to generate SSBs at GTCTCN*N positions. The DNA was then subjected to Sloppymerase that utilized a dNTP mixture lacking dATP. For the samples that would undergo PacBio sequencing, the dNTP mixture was also spiked with biotinylated dUTP to allow the capture of Sloppymerase-modified DNA. Sequencing adapters were attached to the Illumina sample by tagmentation[34]. The PacBio sample was digested with a mixture of restriction enzymes (EcoRI, BamHI, NcoI, and HindIII), followed by the ligation of PCR adapters and pull-down with magnetic streptavidin beads. After a PCR step, which would increase the amount of sample and overwrite deoxyuridine, sequencing adapters were ligated. The sample that would be subjected to Nanopore sequencing was digested with PmlI and PmeI, followed by the ligation of sequencing adapters. To identify SSB sites, we utilized a custom break-detection script that detects a minimum number of consecutive replaced deoxyadenosines

**A**

| 4 dNTPs | GGATCCGGCCAAGCTTCGAGCTGAATTCTGCAGTACATTAATTGGGTTTGGG |
|---|---|
| 1 | GGATCCGGCCAAGCTTCGAGCTGAATTCTGCAGTACATTAATTGGGTTTGGG |
| 2 | GGAACCGGCCAAGCTTCGAGCTGAAATCTGCAGTACATTAATTGG–TTTGGG |
| 3 | GGATCCGACCAAGCTTCGAGCTGATTTCTGCAGTACATTTAATTGGGTTTGGG |
| 4 | GGATCCG–CCAAGCTTCGAGCTGATTCTGCAGTACAGTAATTCGGGTTTGGG |
| 5 | GGAACCGGTCAAGCTTCGAGCTGAATACAGCAGTACAATTAATTGGGTTTGGG |
| 6 | GGACCCGGCCAAGCCTCGACCTGAATTCTGCAGTACATGAAT–GGGTTCGGG |
| 7 | AGATCCGGCCAAACTTCAAGCTGAAT–CAGCAGTACATTAATTGG–––––––– |

A,T,C,G Mutation
A,T,C,G Insertion
– Deletion

| -dATP | GGATCCGGCCAAGCTTCGAGCTGAATTCTGCAGTACATTAATTGGGTTTGGG |
|---|---|
| A1 | GGTTCCGGCCTGGCTTCGTTGCTGGGTTCTGCGGTCCATTAATTGGGTTTGGG |
| A2 | GGC–CCGGCCGGGCTTCG–GCTGGGT–CTGCTGTGCGTTTG––GGGTTTGGG |
| A3 | GGGTCTGGCCGGGCTTCGTGCTGGGTGTCTGCGGTGCT––––TTGGGTGTTGGG |
| A4 | GGGTCCGGCCGGGCTGCGGCCTGGGTTCTGCTGTCCGT–––TTGGGTTTGG– |
| A5 | GGGTCCGGCCGGGCTGCG–GCTTGGGCTCTGCTGTCCGTT–––TGCGTTTGGG |
| A6 | GGGTC–GGCC–GGCTTCGGGCTGGGTTCTGCGGTTCGTT––––GGGTTTGGG |
| A7 | GG–TCCGGCCGTGCTTCG–GCTG–––––TGCCGTCCTTTGGCGCGGGGCCTGGG |

| -dCTP | GGATCCGGCCAAGCTTCGAGCTGAATTCTGCAGTACATTAATTGGGTTTGGG |
|---|---|
| C1 | GGATAAGGAGAAGGTTTGAGTTGAATTGTGTAGTATATTAATTGGGTGTGGG |
| C2 | GGATTGGGGTGAATTTTGG–––––––––––AGTATATTAATTGGGTGTGGG |
| C3 | GGATGAAGTGAA–TTTTGA–––––––––––AGTACATTAATTGGGTTTGGG |
| C4 | GGATAGGGT–AAG–––––AGTAGTAG–––––––TAGTATAATTGGGTTTGGG |
| C5 | GGATTTGAGGTGAA––TTTTGAAGTAAAT––––TAGA––––TTGGGTTTGGG |
| C6 | GGATAAGGGTGAA–ATTTGA–––––––––––AGTATATTAATTGGGTTTGGG |
| C7 | GGATAAGGTTAAG–AGAGAGATGAATTATGAAGTATATTAATTGGGTTTGGG |

**B**

**Fig. 3 | Sanger and Illumina sequencing results for the application of Sloppy-merase to a nicked hairpin. A** Sanger sequencing results when all of the nucleotides were included in the reaction mixture (4 dNTPs) and when either dATP (−dATP) or dCTP (−dCTP) was omitted. The upper row shows the sequence of the hairpin in green letters, with red letters indicating positions where (−dATP or −dCTP) substitutions are expected. Seven clones (1–7, A1–A7, and C1–C7) are shown in the rows below. Substitutions are shown in red, while additional substitutions, insertions, or deletions are marked in blue. **B** Illumina sequencing results with all dNTP (4 dNTPs) and when dATP was omitted (−dATP) from the dNTP mixture. The table shows the percentages of different nucleotides, along with the share of insertions and deletions. The arrow at positions 24/25 indicate the gap where Sloppymerase bind and alters the sequence downstream, position 25–79.

as a signature of Sloppymerase activity. We have herein used a threshold of five consecutive deoxyadenosine replacements to identify an SSB from PacBio reads and Nanopore reads, while three consecutive deoxyadenosine replacements were used to identify an SSB from Illumina reads. The SSB is, hence, located between the last remaining deoxyadenosine and the 5′-end of the first replaced deoxyadenosine. Examples of reads produced by the different sequencing technologies are presented in Fig. 4. The data show that unique Sloppymerase signatures are introduced downstream of Nt.BsmAI nicking sites. In Table 1, we compare the results of detecting SSBs in Nickase-treated *versus* untreated WGS libraries to show that STEEL-seq works reliably on a larger scale. For both Nanopore and Illumina, the rate of predicted SSBs per million reads is roughly an order of magnitude higher in the two Nickase-treated libraries. Also, up to 73% of

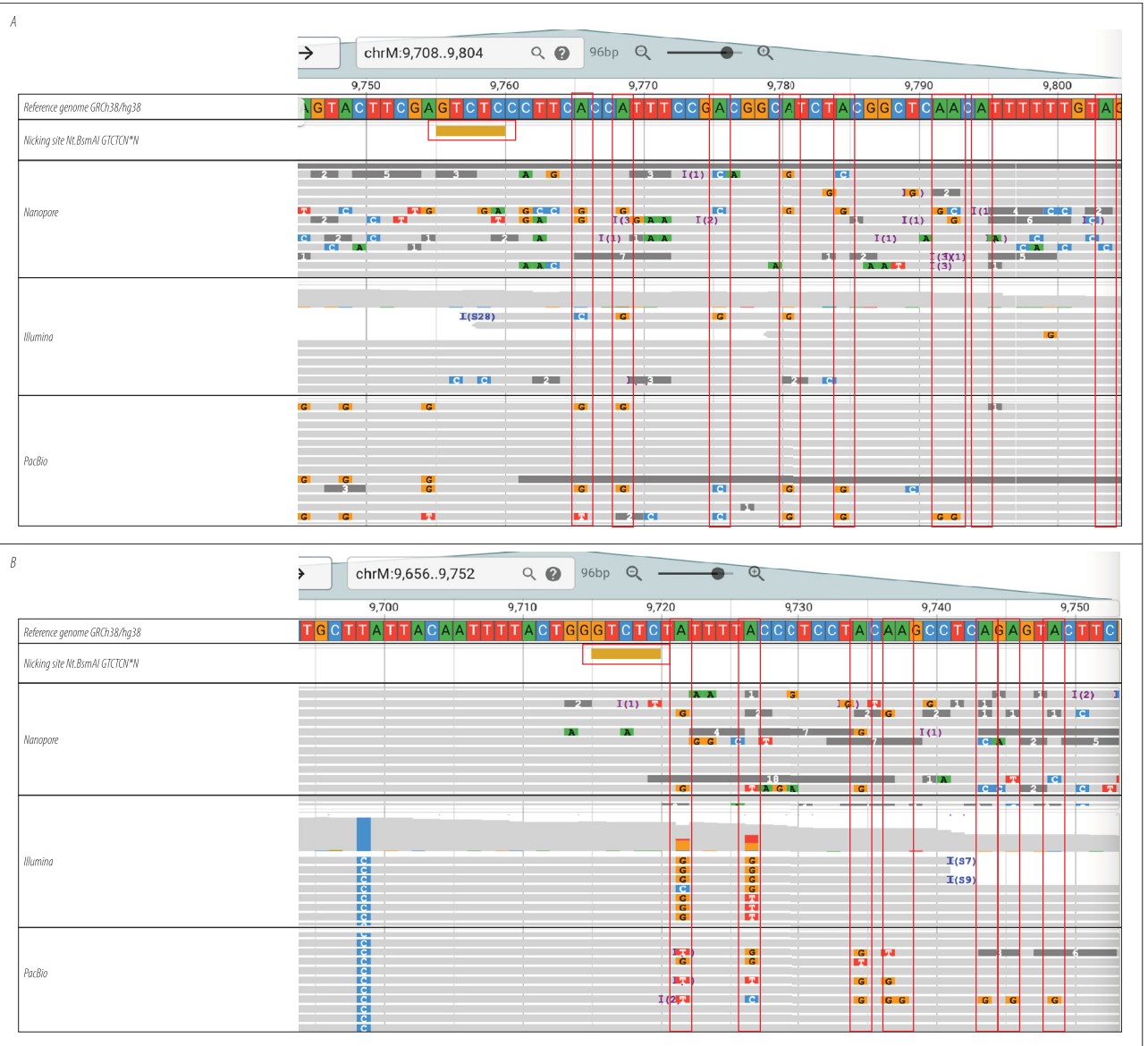

**Fig. 4 | The application of STEEL-seq to identify SSBs.** Testing the utility of STEEL-seq for SSB detection in mitochondrial DNA with different sequencing techniques. STEEL-seq was performed with Nanopore, Illumina, and PacBio whole genome sequencing to detect SSBs at nicking sites. The DNA was nicked with Nt.BsmAI prior to STEEL-seq treatment to create SSBs at GTCTCN*N. The sequencing analysis results show the incorporation of mismatched bases in positions with dA (marked in red) in the reference genome; this was observed downstream of the nicking site at two different loci, **A** chrM:13020 and **B** chrM:14960 on the reverse strand, across all three sequencing technologies. The mitochondrial chromosome was chosen for presentation purposes; coverage on other chromosomes is lower on average. Image based on JBrowse screenshot.

predicted SSBs coincide with nickase target sites. To determine how much overlap would be expected by chance, we generated an annotation track with the same number of nickase sites, but at random locations. This track results in only 5–7% of sites intersecting detected SSBs (not shown in the table). We conclude that STEEL-seq can reliably detect SSBs with high specificity using any of the three sequencing technologies tested.

We then investigated the presence of spontaneous SSBs using STEEL-seq in the human cell lines TK6 and HaCaT (Table 1). Nanopore sequencing of HaCaT cells yields SSB/base pair frequencies of $0.8 \times 10^{-6}$ and $0.7 \times 10^{-6}$, and for Illumina sequencing of TK6 cells, SSB/ base pair frequencies of $2.7 \times 10^{-6}$ and $3.8 \times 10^{-6}$. The results demonstrate an enrichment of SSBs in regions proximal to transcription start sites (TSS) (Fig. 5A), consistent with findings reported using SSiNGLe to identify SSBs in the human genome[35]. To determine whether the presence of SSBs in promoter regions correlates with transcription, we

conducted both transcriptome analysis and SSB profiling. HaCaT cells were serum-starved overnight to minimize replication and transcriptional noise. The cells were then preincubated for 1.5 h with 3 μM BMH-21 or DMSO. BMH-21 is a DNA intercalator capable of interfering with transcription by inducing degradation of RNA polymerase I and II[36–39]. The cells were then stimulated with 5 ng/ml TGFβ to induce the expression of TGFβ-SMAD target genes, or not, after which the cells were harvested and RNA and DNA were extracted. STEEL-seq was used to determine SSB frequency and to map SSB locations, and RNA sequencing and qRT-PCR were performed to determine gene expression (Fig. 5B). The SSBs/base pair frequency at a whole genome level was determined to be $1.7 \times 10^{-6}$ for the DMSO control, which was increased to $2.9 \times 10^{-6}$ for the TGFβ-stimulated sample. For the BMH-21-treated sample, the SSB/base pair frequency was $1.7 \times 10^{-6}$ and for the BMH-21 + TGFβ-stimulated sample $2.1 \times 10^{-6}$. To confirm the expression of TGFβ-induced genes, a panel of well-characterized target

**Table 1 | Overview of the analyzed Sloppymerase-treated WGS libraries from all three sequencing technologies**

| Accession | Cell line (Treatment) | SSBs induced by | Technology | Aligned nucleotides (billions) | Detected unique SSBs | SSBs per million base pairs | SSBs coinciding w/ nickase site |
|---|---|---|---|---|---|---|---|
| ERR13611927 | HaCaT | Nickase | Nanopore | 8.3 | 117,950 | 14.3 | 59.5% |
| ERR13611929 | HaCaT | Nickase | Nanopore | 9.3 | 104,262 | 11.2 | 62.2% |
| ERR13611926 | HaCaT | – | Nanopore | 21.7 | 14,925 | 0.7 | 7.6% |
| ERR13611928 | HaCaT | – | Nanopore | 16.8 | 13,721 | 0.8 | 8.4% |
| ERR13611931 | TK6 | Nickase | PacBio HiFi | 0.2 | 8131 | 49.3 | 73.0% |
| ERR13611934 | TK6 | Nickase | Illumina | 6.7 | 40,097 | 6.0 | 49.7% |
| ERR13611935 | TK6 | Nickase | Illumina | 7.0 | 51,883 | 7.5 | 52.4% |
| ERR13611936 | TK6 | – | Illumina | 8.2 | 22,074 | 2.7 | 6.6% |
| ERR13611937 | TK6 | – | Illumina | 9.7 | 37,141 | 3.8 | 6.4% |
| ERR15076661 | HaCaT (DMSO) | – | Illumina | 122.9 | 213,008 | 1.7 | 8.4% |
| ERR15076662 | HaCaT (TGFβ) | – | Illumina | 186.5 | 534,812 | 2.9 | 8.1% |
| ERR15076663 | HaCaT (BMH-21) | – | Illumina | 130.1 | 216,583 | 1.7 | 8.2% |
| ERR15076664 | HaCaT (BMH-21 + TGFβ) | – | Illumina | 190.7 | 396,605 | 2.1 | 8.6% |

Alignments are considered usable if they are primary (i.e., split alignments of long reads are excluded) and have at most 10% errors. SSBs were detected using the script and parameters as described in the Methods section. A detected break is considered to coincide with a Nt.BsmAI nickase site if its distance to the nickase motif is at most 10 bp (using bedtools -w 10).

genes was analyzed by qRT-PCR. TGFβ-stimulation increased the expression of typical early response genes; i.e., *SERPINE 1, IL11, JUNB,* and *SMAD7*; whereas our control gene, which was not expected to be induced, *CHD4*, was unaffected by TGFβ-stimulation. BMH-21 was found to reduce the TGFβ-induced expression to varying extents. We then determined the genomic regions in which the SSBs were located and observed enrichment at the promoter region (Fig. 5C, first row). To determine if the SSBs correlate with gene expression, we selected genes that were not detected in any of the sequenced transcriptomes (15,598 genes) and determined the SSB/base pair frequency at the different genomic regions. We observed a reduction in SSB/base pair frequency: $0.9 \times 10^{-6}$ for DMSO, $1.6 \times 10^{-6}$ for TGFβ, $0.9 \times 10^{-6}$ for BMH-21 and $1.1 \times 10^{-6}$ for BMH-21 + TGFβ (Fig. 5C, second row). Next, we analyzed the SSB/base pair frequency at the different genomic regions for the 60 genes that were upregulated upon TGFβ-stimulation (Supplementary Table 1) and observed an increase in SSB/base pair frequency $2.4 \times 10^{-6}$ for DMSO, $4.1 \times 10^{-6}$ for TGFβ, $2.8 \times 10^{-6}$ for BMH-21, and $3.1 \times 10^{-6}$ for BMH-21 + TGFβ (Fig. 5C, third row). The SSB/base pair frequency was lowest in the exons and highest in 5′ UTR and 1st kb of the promoter region. TGFβ-stimulation increased SSB/base pair frequency, in both DMSO and BMH-21 treated samples.

## Discussion

We do not yet have a complete understanding of exactly when and where SSBs form in the genome, and how SSBs contribute to the introduction of mutations which can cause cancer. Studies of SSBs have been hampered by the lack of specific methods. However, recent years have been characterized by significant developments in several new methods for the precise detection of SSBs. Methods such as SSB-seq, in which the region downstream of a SSB is modified by the incorporation of biotinylated nucleotides, can pull down modified regions. However, this technique does not allow researchers to pinpoint exactly where in the DNA the break occurred. Methods such as SSiNGLe, Nick-seq, and GLOE-seq provide information on the positions of free 3′-OH ends, but cannot distinguish if these were the result of an SSB or DSB. By labeling SSBs via a template-specific DNA polymerase instead of TdT, STEEL-seq provides sequence information both upstream and downstream of SSBs to enable the identification of bona fide SSBs. This required the engineering of a highly error-prone DNA polymerase, which we describe in detail in this paper. Another advantage of the described approach is that the random insertion of mismatches will label each molecule with a unique molecular signature, which will allow researchers to determine whether multiple sequencing reads originate from different cells.

We performed whole genome sequencing, using STEEL-seq, enabling direct measurement of SSB/base pair frequency. To accomplish this, we did not enrich for Sloppymerase-modified DNA fragments. Instead, we directly proceeded with the generation of sequencing libraries. Hence, only a low fraction of the reads contained Sloppymerase signatures that identified SSBs. Previous studies using the SSiNGLe method have reported increased SSBs in promoter regions. An increase in SSBs was observed in regulatory elements, introns, and exons[13] and the SSBs were found to be enriched around TSS[35]. Although DSBs may also be recorded using this method, the frequency should be so low that it does not change the result significantly. The SSiNGLe method relies on the enrichment of labelled fragments, and provides precise determination on positions where SSBs occur and relative abundance. Although STEEL-seq cannot be used to exactly map where the SSB is located, it still provides very high resolution, within a few nucleotides of the SSB position, i.e., the region between last remaining A and first missing A. As STEEL-seq modifies the DNA, without the need for fragmentation it allows any downstream protocol, it is in that aspect similar to bisulfite conversion for analysis of DNA methylation status[40], and may possibly be combined with methods such as BLESS or BLISS[12] for simultaneous quantification of both SSBs and DSBs. The sequencing method can also easily be modified to determine the frequency of damaged bases (e.g., cis-syn thymine dimers or 8-oxoG). By utilizing enzymes that remove these structures, such as T4 pyrimidine DNA glycosylase (T4 PDG) or Formamidopyrimidine DNA Glycosylase (FPG), ssDNA breaks will be created. These ssDNA breaks can then be labeled with Sloppymerase. By omitting another nucleotide, e.g., dCTP, different labels will be produced that may allow researchers to simultaneously monitor multiple types of DNA damage. As Sloppymerase allows for labelling in fixed and permeabilized cells (Supplementary Fig. 5), it should be possible to perform STEEL-seq prior to DNA extraction to facilitate single cell analysis.

Using STEEL-seq, we were able to directly quantify SSB/base pair frequency in the human genome, yielding similar results across different sequencing platforms and cell lines. The frequency was in the

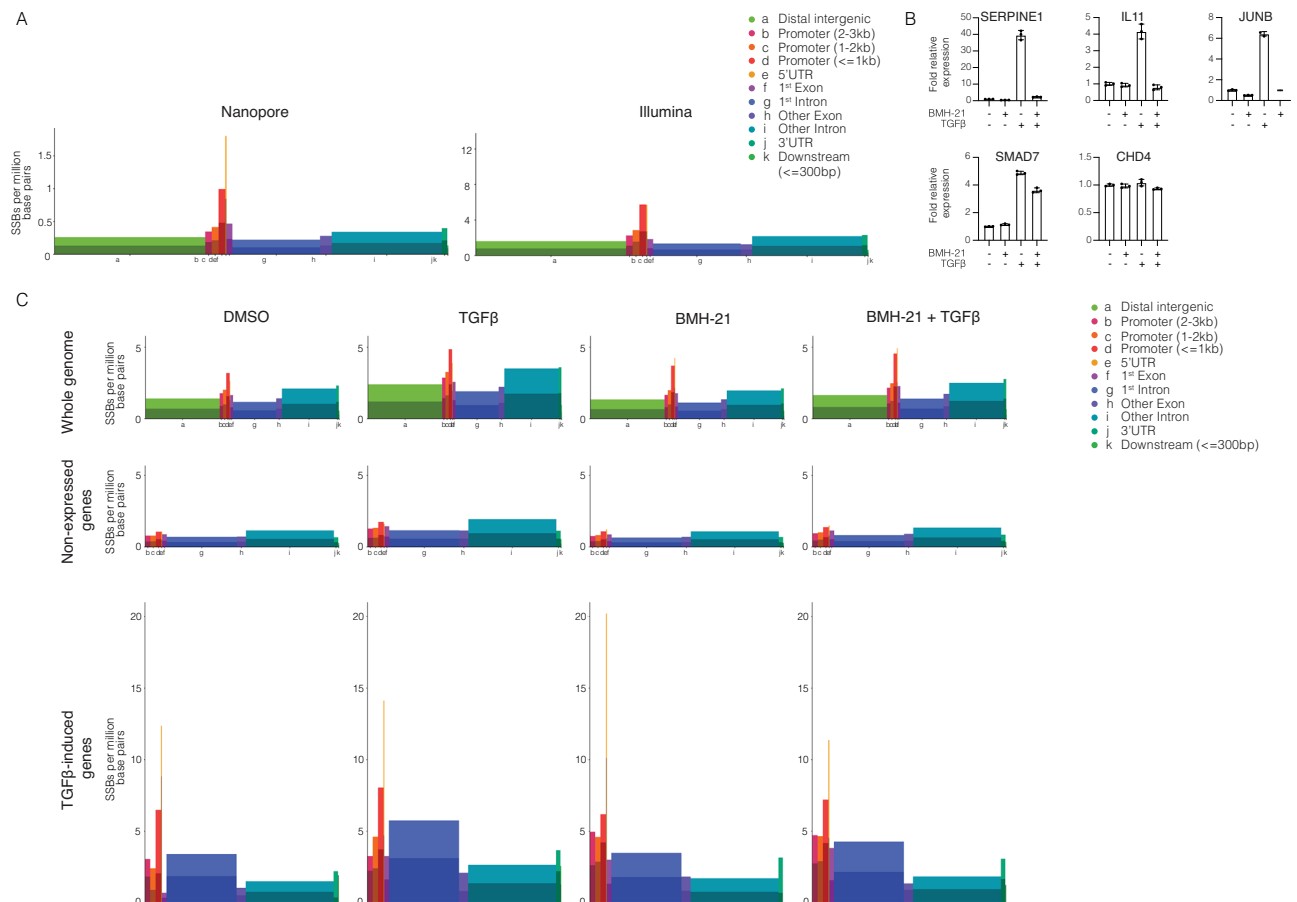

**Fig. 5 | Detection of SSBs using Nanopore and Illumina sequencing. A** Whole genome analysis of STEEL-seq of HaCaT cells, using Nanopore, and TK6 cells with Illumina. The figure shows SSB/base pair frequency for different regions. The width of the regions is proportional to the regions size in the genome. Lighter hue of the bars represents SSBs on the DNA strand in the forward direction, in relation to gene orientation, and darker hue represents SSBs on the DNA strand in the reverse orientation. **B** qRT-PCR analysis to confirm that TGFβ-stimulation induced expression of target genes was performed as a control before DNA samples were used for STEEL-seq. HaCaT cells were treated with DMSO or BMH-21 for 3 h, and stimulated or not with 5 ng/ml TGFβ for the last 1.5 h. Data are presented as mean values +/− SD, technical replicate $n = 3$. TGFβ-induced expression of target genes was confirmed by RNA sequencing (see Supplementary Table 1). **C** STEEL-seq results, using Illumina, of HaCaT cells treated with DMSO or BMH-21, stimulated or not with TGFβ. Upper row represent whole genome analysis. Second row represent selected genes where no transcript could be found. Third row show genes induced by TGFβ-treatment. The figure shows SSB/base pair frequency for different regions. The width of the regions is proportional to the regions size in the genome. Lighter hue of the bars represents SSBs on the DNA strand in the forward direction, in relation to gene orientation, and darker hue represents SSBs on the DNA strand in the reverse orientation.

range between 0.7 and $3.8 \times 10^{-6}$, which is similar to the frequency reported for *E. coli*, $1.2 \times 10^{-6}$, using the RADAR-seq method for detection of SSBs[10]. We observed that the SSB/base pair frequency was reduced in non-expressed genes, while it was higher in active genes and was further increased upon growth factor stimulation. TGFβ-stimulation resulted in increased expression of target genes and an increase in SSB/base pair frequency, which was partially inhibited by BMH-21. The concentration of BMH-21 was chosen to only partially induce Z-DNA formation[38], thereby reducing any potential toxic effects. Further experiments are required to determine the effects at higher concentrations.

When analyzing the SSB/base pair frequency at the whole genome scale, we observe a consistent profile, where SSB/base pair are enriched at certain regions 5′UTR > promoter region > 3′UTR > later introns > distal intergenic regions > exons > 1st intron but for active genes there was an enrichment in regions close to TSS and 1st intron, together with a relative reduction in exons and later introns. Both SSiNGLe and STEEL-seq detect SSBs with 3′-OH termini and find an enrichment around TSS. As Topoisomerase I (TOP1), produces 3′-phosphate termini[41], such SSBs will not allow extension by DNA polymerases, and will therefore not be detected by STEEL-seq or SSiNGLe. For analysis of SSBs generated by

TOP1, the DNA could be treated with e.g., Shrimp alkaline phosphatase, Endonuclease IV or APE1 to convert 3′-phosphate into 3′-OH termini, prior to STEEL-seq. The enzyme responsible for inducing the detected SSBs is still not identified, but the methods to detect SSBs provide tools to investigate SSB formation and role in regulating gene expression. A topoisomerase that generates SSBs with 3′-OH termini is Topoisomerase IIIα (TOP3A)[42]. Other possible enzymes are the endonucleases XPG and XPF that are recruited by the general transcription factor TFIIH[43]. Activation of gene expression requires chromatin remodeling and recruitment of co-activator complexes, the mediator complex, the general transcription factors, and RNA polymerase II. The general transcription factor TFIID binds the TATA box, recruits RNA polymerase II, and the other general transcription factors (including TFIIH). TFIIH opens the promoter (through its helicases XPB and XPD) and activates RNA polymerase II by phosphorylating its carboxy-terminal tail. TFIIH recruits XPG and XPF, which enhances the helicase activity of TFIIH[44]. Interestingly, apart from its function in transcription, TFIIH is responsible for excising damaged DNA in Nucleotide Excision Repair. DNA lesions are detected by XPC causing distortion of the DNA helix that allows the recruitment of TFIIH. TFIIH opens up the DNA spanning the DNA lesion and recruits XPG and XPF that cut DNA on both sides of the lesion, XPF at

the 5′ side and XPG at the 3′ side of the lesion[43]. This indicates that XPG and XPF may also be responsible for introducing SSBs at the promoter region, which might in part explain how they enhance the helicase activity of TFIIH. The binding of transcription factors may distort the DNA helix to facilitate the recruitment of TFIIH, and the subsequent recruitment of XPG and XPF to the promoter regions, including 5′UTR. The intriguing finding that SSBs are enriched at regions flanking TSS and that the frequency of SSB/base pair are higher in active genes suggests that there is more to discover regarding the mechanisms of gene expression. Methods such as STEEL-seq, SSiNGLe among others, provide the necessary tools for revealing the roles of SSBs.

We have herein shown that our engineered chimeric DNA polymerase, Sloppymerase, can replicate DNA when one, or even two, nucleotides are missing. As Sloppymerase is equipped with a domain that possesses 5′–3′ exonuclease activity, it will degrade the DNA strand in front of the DNA polymerase, which allows it to exchange nucleotides over a longer stretch. We have also shown that Sloppymerase can be used to selectively label SSBs. The STEEL-seq method might also work with other highly error-prone, catalytic dead DNA polymerases. The aim of the project was, in essence, to develop the worst DNA polymerase that has ever existed to enable the robust development of STEEL-seq. The properties of Sloppymerase may, in addition to STEEL-seq, be exploited in several types of biotechnological applications.

## Methods

### Production of Sloppymerase

To purify Sloppymerase, an overnight culture was created from BL21 *E. coli* (Thermo Fisher Scientific) that have been transformed by heat-shock with the Sloppymerase vector (VectorBuilder). The bacteria were inoculated with 4 ml lysogeny broth (LB) containing 100 µg/ml ampicillin (Merck) at 37 °C with 225 rpm. The next day the overnight culture was transferred to a 1 L flask with 200 ml LB containing 100 µg/ml ampicillin. The culture was grown at 37 °C with 225 rpm to an OD600 of 0.5. The expression of Sloppymerase was induced with 0.2% L-arabinose (Merck) and allowed to grow at 16 °C overnight with 225 rpm. The next day the cells were harvested in a Sorwall centrifuge for 15 min at 6000 × *g* at 4 °C in a GSA rotor. After centrifugation the cells were lysed with 20 ml binding buffer (50 mM sodium phosphate, 500 mM NaCl, pH 7.4, 1 mM MgCl₂, 0.25 % TritonX-100, 1× complete protease inhibitor EDTA free (Merck), 0.2 mg/ml lysozyme (Thermo Fisher Scientific)) and incubated for 30 min at 4 °C. To clear the lysate, the samples were centrifuged for 15 min at 13,000 × *g* at 4 °C and passed through a 0.45 µm Filtropur S syringe filter (Sarstedt). The His GraviTrap ™ Talon® (Cytivia) were equilibrated according to the producer's manual before loading the samples. The columns were washed with 3 × 10 ml washing buffer (50 mM sodium phosphate, 500 mM NaCl, 5 mM imidazole, pH 7.4) and thereafter the samples were eluted with elution buffer (50 mM sodium phosphate, 500 mM NaCl, 50 mM imidazole pH 7.4). To exchange the buffer to the storage buffer (25 mM Tris-HCl, 1 mM DL-Dithiothreitol (DTT), 0.2 mM EDTA (Merck), pH 7.4), the enzyme was concentrated down with Amicon® 10 kDa Ultra Centrifugal filters (Merck) and reconstituted to the 2× sample volume with 2× storage buffer. This was repeated twice before the concentration was measured with nanodrop and glycerol (Merck) was added to a final concentration of 50%. To check the purity of the enzyme, samples from the purification were run on a NuPAGE Novex 4–12% Bis-Tris gel and thereafter stained with Coomassie Brilliant Blue G-250 Dye (Thermo Fisher Scientific) for 30 min. After destaining with water, the gel was scanned at 700 nm with an Odyssey® Fc imaging system (Li-Cor).

### LC-MS analysis

To confirm the amino acid sequence of the generated enzyme, a sample was prepared for LC-MS analysis. For filter-aided sample preparation, 20 µg of protein lysate were placed on the filter unit (Microcon-30 kDa; Merck, Darmstadt, Germany) and washed with a buffer containing 8 M urea and 100 mM Tris (pH 8.5). First 8 mM DTT were added followed by an incubation at 56 °C for 15 min and then 50 mM of IAA. After 20 min of incubating at room temperature, excess IAA was removed with 8 mM DTT (incubation at 56 °C for 15 min). After each incubation, the sample was washed twice with Tris buffer. Finally, washing with NH₄HCO₃ was performed, trypsin was added [enzyme–protein ratio 1:50 (w/w)] and the samples were placed in a wet chamber at 37 °C. After incubation overnight, the resulting peptides were washed from the filter by adding 50 mM NH₄HCO₃ and centrifuging at 14,000 × *g* for 10 min twice. Trifluoroacetic acid [final concentration of 1% (v/v)] was added, the samples were dried and reconstituted in a solution containing 3% acetonitrile and 0.1% formic acid in water to a final concentration of 150 ng protein/µL.

For tryptic peptide analysis, a nanoAcquity UPLC system equipped with a C18, 5 µm, 180 µm × 20 mm trap column and a HSS-T3 C18 1.8 µm, 75 µm × 100 mm analytical column (Waters Corporation, Manchester, UK) was coupled to a Synapt G2 Si HDMS mass spectrometer with an electrospray ionization source (Waters Corporation, Manchester, UK). Mobile phase A contained 0.1% formic acid and 3% dimethyl sulfoxide in water and mobile phase B 0.1% formic acid and 3% dimethyl sulfoxide in acetonitrile. 300 ng of protein was injected in trapping mode. The peptides were separated at 40 °C with a gradient run from 3 to 40% (v/v) mobile phase B at a flow rate of 0.3 µl/min over 120 min. Via the reference channel, a lock mass solution composed of [Glu1]-fibrinopeptide B (0.1 µM) and leu-enkephalin (1 µM) was introduced every 60 s. Peptide analysis was performed in positive ionization mode using the ultra-definition MSE approach. The reproducibility and stability of the method were controlled with a commercially available HeLa digest (Thermo Scientific, Waltham, MA). ProteinLynx Global Server (PLGS) (version 3.0.3, Waters Corporation, Milford, MA) and FragPipe (https://github.com/Nesvilab/FragPipe/) was used for data processing. Peak picking was conducted with PLGS and the resulting feature list was searched with FragPipe against an *E. coli* FASTA-file with the addition of the amino-acid sequence for the engineered enzyme. Search parameters included trypsin as the digest reagent, one missed cleavage was allowed and the FDR limit was set to 0.01.

### Sloppymerase activity

To test the activity of the purified enzyme, an incomplete hairpin was used. Two oligonucleotides (hairpin A1 and A2, (Integrated DNA Technologies)) were ligated together by mixing 20 µM of each oligonucleotide and ligate the oligos for 48 h at 4 °C end-over-end in 1× T4 DNA ligation buffer (50 mM Tris-HCl (pH 7.6), 10 mM MgCl₂, 1 mM ATP, 1 mM DTT, 5% (w/v) polyethylene glycol-8000) and 0.1 U/µl T4 DNA ligase (Thermo Fisher Scientific). The reaction was heat-inactivated at 65 °C for 20 min. The incomplete hairpin was hybridized to a complementary oligonucleotide (hairpin A3) to create a nicked hairpin. 0.02 µM of the hairpin was then treated with Sloppymerase with either all four dNTPs: (0.1 mM dATP, 0.1 mM dCTP, 0.1 mM dGTP, 0.05 mM dTTP (Thermo Fisher Scientific) and 0.05 mM Biotin-11-dUTP (Jena Bioscience)) or omitting either dATP or dCTP in 1× NEBuffer 2.1 (New England Biolabs) with 0.1 mM MnCl₂ (Merck) and 0.035 µg/µl Sloppymerase. The samples were incubated at 37 °C for 60 min unless stated otherwise and then heat-inactivated at 75 °C for 20 min. The samples were mixed to a final concentration of 1× with Novex™ TBE-Urea Sample Buffer (2×) (Thermo Fisher Scientific) and heated up to 95 °C for 5 min before running the samples on Novex™ TBE-Urea gels, 10% (Thermo Fisher Scientific) at denaturing conditions to evaluate DNA polymerase and 5′–3′ exonuclease activity. The gel was stained with SYBR™ Gold nucleic acid gel stain (Thermo Fisher Scientific) and scanned at 600 nm with an Odyssey® Fc imaging system (Li-Cor). For evaluating if Sloppymerase can incorporate biotin-dUTP, the gel was additionally stained with IRDye® 800CW Streptavidin (926-

32230. LI-COR) in a final concentration of 0.2 μg/ml before scanning the gel at 800 nm. The sequence of the oligonucleotides used in this experiment is presented in Table 2, oligo no 1–3.

## Sanger sequencing of Sloppymerase-treated Hairpin

For the Sanger sequencing, 20 μM of hairpin S1, S2, and S3 were mixed together to create a nicked hairpin. The oligonucleotides were ligated overnight as described above. 0.02 μM of the hairpin was then treated with Sloppymerase with either all four dNTPs: (0.1 mM dATP, 0.1 mM dCTP, 0.1 mM dGTP, and 0.1 mM dTTP or omitting either dATP or dCTP in 1× NEBuffer 2.1 with 0.1 mM MnCl₂ and 0.035 μg/μl Sloppymerase. The samples were incubated at 37 °C for 60 min and thereafter heat-inactivated for 20 min at 75 °C. To prepare the sequences for PCR, adapter S1 was ligated overnight at 4 °C with 1 mM ATP, 0.1 U/μl T4 DNA Ligase, and 0.3 U/μl T4 Polynucleotide Kinase (New England Biolabs).

For the PCR 0.04 μM of treated hairpin was added to 1× Phusion Green HF buffer (Thermo Fisher Scientific), 0.2 mM dNTPs (Thermo Fisher Scientific), 0.5 μM of forward and reverse primers S1 with 0.02 U/μl of Phusion U Hot Start DNA Polymerase (Thermo Fisher Scientific). The hairpin was amplified with a thermocycler for 7 s at 98 °C, 20 s at 60 °C, and 20 s at 72 °C for 20 cycles. The PCR product was cloned with Zero Blunt™ TOPO™ PCR Cloning Kit (Thermo Fisher Scientific) for sequencing according to the manufacturer's manual. Another PCR was performed with the cloning products by dipping a pipette tip in a single colony and then in a tube with master mix consisting of 1× Platinum II HS buffer, 0.2 mM dNTPs, 0.04 U/μl Platinum II HS polymerase (Thermo Fisher Scientific) and 0.2 μM forward and reverse primers S2. The reaction was amplified with a thermocycler for 15 s at 94 °C, 15 s at 60 °C, and 15 s at 68 °C for 30 cycles. The PCR products were cleaned up with 0.8 U/μl exonuclease I and 0.03 U/μl shrimp alkaline phosphatase (New England Biolabs) for 30 min at 37 °C and heat-inactivated at 80 °C for 20 min. The samples were then prepared for sequencing according to the platform's instructions for TubeSeq Service (Eurofins Genomics). The samples were sequenced on ABI 3730XL sequencing machines using cycle sequencing technology (dideoxy chain termination/cycle sequencing). The sequence of the oligonucleotides used in this experiment is presented in Table 2, oligo no 4–11.

## Illumina sequencing of Sloppymerase-treated Hairpin

To test the activity of Sloppymerase on hairpins with Illumina sequencing, hairpins prepared and treated with Sloppymerase as described above for 120 min. Thereafter, the elongated hairpins were treated with 0.4 U/L BamHI-HF (NEB, R3136L) for 2 h at 37 °C to remove the loop of the hairpin. The buffer was exchanged for T4 ligation buffer using Zeba™ Spin Columns as described previously, and adapters were ligated (equimolar) to the truncated hairpin with 0.15 U/L T4 DNA ligase for 16 h in intervals of 30 min, cycling between 37 °C and 4 °C. For the PCR, 0.04 μM of treated hairpin was added to 1× Phusion Hot Start II High-Fidelity PCR Master Mix (F566S, Thermo Fisher Scientific) along with 0.5 μM of forward and reverse primers. The hairpin was amplified using a thermocycler (15 s at 98 °C, 15 s at 64 °C, and 10 s at 72 °C for 16 cycles; with 30 s of activation at 98 °C before the cycles and 10 min at 72 °C afterwards). The samples were run on Novex™ TBE Gels, 6% (Thermo Fisher Scientific) for quality control, followed by PCR cleanup with the MinElute PCR Purification Kit (Qiagen) according to the manufacturer's protocol, and eluted in nuclease-free water. The samples were then diluted and prepared according to the sequencing platform's requirements and were sequenced with Illumina. SNP&SEQ Technology Platform performed 75 cycles paired-end sequencing in a MiSeq Nano v2 flowcell. The sequence of the oligonucleotides used in this experiment is presented in Table 2, oligo no 12–22.

## Cell culture

Human lymphoblastoid TK6 cells (ATCC, CRL-8015) were cultivated in RPMI 1640 medium supplemented with 10% fetal bovine serum (FBS),

and Human keratinocytes (HaCaT, 3000493, CLS) were cultivated in DMEM+Glutamax supplemented with 10% FBS (culture medium, all from Thermofisher Scientific) at an atmosphere-controlled environment of 37 °C and 5% CO₂. For the starvation medium, the amount of FBS was reduced to 0.2% FBS.

## STEEL-seq using Illumina sequencing

For Illumina sequencing, DNA has been extracted from TK-6 lymphoblast cells. The DNA was extracted with DNeasy Blood & Tissue Kit (Qiagen) according to the manufacturer's manual. 100 ng of the extracted WT DNA was additionally incubated with 20 U of Nt.BsmAI (New England Biolabs) and 1× Cutsmart buffer (New England Biolabs) to introduce nicks. After a buffer exchange to ddH₂O with Zeba™ Spin Desalting Columns, 7 K MWCO, 0.5 ml (Thermo Fisher Scientific), the samples were incubated with 1× NEBuffer 2.1, a mixture of dNTPs (0.1 mM dCTP, 0.2 mM dGTP, 0.1 mM dTTP), 0.1 mM MnCl₂ (Merck), and 0.035 μg/μl Sloppymerase for 120 min at 37 °C. Thereafter, the samples were heat-inactivated for 20 min at 75 °C. For tagmentation, 10 ng of purified DNA/sample was added to a transposase mixture with 12.5 μL 2× TD buffer, 1.25 μL Tn5 transposase (2 μM), produced and assembled according to Picelli et al.[34] and ddH₂O to bring the reaction volume up to 50 μL. The Tn5 adapters used were: I3, I4, and I5. The transposase mixture was incubated at 55 °C for 7 min and purified using the QIAquick PCR cleanup kit (QIAGEN), following manufacturer's protocol and eluted in 20 μL ddH₂O. The purified DNA was added to 25 μL NEBNext High-Fidelity 2× PCR Master Mix (New England Biolabs), 2.5 μL of forward primer: I1 (25 nM) and 2.5 μL of reverse primer: I1, I2, or I3 (25 nM). It was amplified with a thermocycler following 72 °C for 5 min, 98 °C for 30 s, and thermocycling at 98 °C for 10 s, 63 °C for 30 s, and 72 °C for 1 min for 9 cycles. After amplification, the samples were purified using SPRIselect beads (Beckman) at a 1:1 ratio, following the manufacturer's protocol for the Left workflow. The purified samples were analyzed with the 2200 TapeStation System (Agilent) and kept at −20 °C for sequencing. The samples were then sequenced with Illumina NovaSeq 6000. SNP&SEQ Technology Platform performed 150 cycles paired-end sequencing in one lane of a SP flowcell. The sequence of the oligonucleotides used in this experiment is presented in Table 2, oligo no 23–29.

## Comparing gene expression to detected SSBs with STEEL-seq using Illumina sequencing

HaCaT cells were seeded in culture medium to confluence and left to adhere overnight. The next day, the cells were starved overnight in starvation medium and subsequently preincubated with DMSO or 3 μM BMH-21 (Sigma-Aldrich, SML1183) for 1.5 h. They were then treated with medium or 5 ng/ml TGFβ−1 (PeproTech) for an additional 1.5 h. Total RNA was extracted using the RNeasy Mini Kit (74106, Qiagen), and DNase treatment was performed with 1 U/μl DNase I (Thermofisher Scientific), both according to the manufacturer's instructions. The DNA was extracted with the DNeasy Blood & Tissue Kit (Qiagen) according to the manufacturer's instructions. The extracted DNA was then treated for 60 min at 37 °C with 0,2 μg/μl Sloppymerase and a mixture of dNTPs (0.1 mM dCTP, 0.2 mM dGTP, 0.1 mM dTTP) and thereafter heat-inactivated for 20 min at 65 °C. To seal any gaps, the samples were treated with T4 DNA ligase and before being heat-inactivated. Instead of a sonication step, the samples were digested with 20 U RsaI and 40 U PmlI to create blunt ends for 45 min at 37 °C. The samples were prepared for sequencing with the MinElute PCR purification kit (Qiagen) according to the manufacturer's instructions and eluted in EB buffer (10 mM Tris-Cl, pH 8.5). The samples were then sequenced with Illumina NovaSeq 6000. Sequencing was performed by the SNP&SEQ Technology Platform using a NovaSeq X Plus system, 10B flow cell and XLEAP-SBS sequencing chemistry with 150 bp paired-end reads. For qRT-PCR analysis, cDNA samples were diluted tenfold with water. The qRT-PCR was conducted

**Table 2 | Oligonucleotide sequences used in the experiments**

| Oligo no | Name | Sequence 5' to 3' |
|---|---|---|
| 1 | Hairpin A1 | CCCAAACCCAATTAATGTACTGCAGAATTCAGCTCGAAGCTTGGCCGGATCCGTGAGCTGTCGTC |
| 2 | Hairpin A2 | /5Phos/TCAGATCGGATACGGCGACCACCGAGATCTACACCCTGCGGGACACTCTTTCCCTACACGACGCTCTTCCGATCTGAGACGACAGCTCAC |
| 3 | Hairpin A3 | CCGGCCAAGCTTCGAGCTGAATTCTGCAGTACATTAATTGGGTTTGGG |
| 4 | Hairpin S1 | CCCAAACCCAATTAATGTACTGCAGAATTCAGCTCGAAGCTTGGCCGGATCCAGCGTGGGACTGAGTC |
| 5 | Hairpin S2 | /5Phos/GTCTCGTGTCTGTAAAAACGTACGTAGATGCCATTTCTAAAAAAACAGACACGAGACGACTCAGTCCCACGCT |
| 6 | Hairpin S3 | CCGGCCAAGCTTCGAGCTGAATTCTGCAGTACATTAATTGGGTTTGGG |
| 7 | Adapter S1 | /5Phos/CGCACTGAGACTGATATGTGAAAAATTAGATTGGATAACTGCGCAGAAAAACACATATCAGTCTCAGTGCG |
| 8 | Fwd primer S1 | CTGCGCAGTTATCCAATCTAA |
| 9 | Rev primer S1 | CGTACGTAGATGCCATTTCTA |
| 10 | Fwd primer S2 | GTAAAACGACGGCCAG |
| 11 | Rev primer S2 | CAGGAAACAGCTATGAC |
| 12 | Hairpin (sequencing) | GCAACCAGACAGGTATGAACGCGCGCTACTGTTTTCAGCTCGAAGCTTGGCTGTGGAGACCGTGGGACTGAGTC |
| 13 | Hairpin (sequencing) | /5Phos/GGATCCCGTGTCTGTCATGTTACAGACACGGGATCCGACTCAGTCCCACGGTCTC |
| 14 | Hairpin (sequencing) | CAGCCAAGCTTCGAGCTGAAAACAGTAGCGCGCGTTCATACCTGTCTGGTTG/3ddC/ |
| 15 | Adapter i5-1F | /5Phos/AATGATACGGCGACCACCGAGATCTACACAGCGCTAGACACTCTTTCCCTACACGACGCTCTTCCGATCT |
| 16 | Adapter i5-1R | /5Phos/GATCAGATCGGAAGAGCGTCGTGTAGGGAAAGAGTGT/3ddC/ |
| 17 | Adapter i7-1F | /5Phos/GATCGGAAGAGCACACGTCTGAACTCCAGTCACCCGCGGTTATCTCGTATGCCGTCTTCTGCTTG |
| 18 | Adapter i7-1R | /5Phos/GTGACTGGAGTTCAGACGTGTGCTCTTCCGATC |
| 19 | Adapter i5-2F | /5Phos/AATGATACGGCGACCACCGAGATCTACACCGCAGACGACACTCTTTCCCTACACGACGCTCTTCCGATCT |
| 20 | Adapter i7-2F | /5Phos/GATCGGAAGAGCACACGTCTGAACTCCAGTCACGGACTTGGATCTCGTATGCCGTCTTCTGCTTG |
| 21 | Primer FWD | AATGATACGGCGACCACCGA |
| 22 | Primer REV | CAAGCAGAAGACGGCATACG |
| 23 | Adapter I3 | TCGTCGGCAGCGTCAGATGTGTATAAGAGACAG |
| 24 | Adapter I4 | GTCTCGTGGGCTCGGAGATGTGTATAAGAGACAG |
| 25 | Adapter I5 | CTGTCTCTTATACACATCT |
| 26 | Fwd primer I1 | AATGATACGGCGACCACCGAGATCTACACCCTGCGGGTCGTCGGCAGCGTCAGATGTGTAT |
| 27 | Rev primer I1 | CAAGCAGAAGACGGCATACGAGATCTGTATTTGTCTCGTGGGCTCGGAGATGTG |
| 28 | Rev primer I2 | CAAGCAGAAGACGGCATACGAGATGACCCAAGGTCTCGTGGGCTCGGAGATGTG |
| 29 | Rev primer I3 | CAAGCAGAAGACGGCATACGAGATTTTAACGCGTCTCGTGGGCTCGGAGATGTG |
| 30 | CHD4 FW | CTGTTGCTGACTGGGACACCAT |
| 31 | CHD4 Rev | TGGTCCTCCTTGGCAATGTCAG |
| 32 | GAPDH FW | GGAGTCAACGGATTTGGTCGTA |
| 33 | GAPDH Rev | GGCAACAATATCCACTTTACCA |
| 34 | IL-11 FW | CGAGCGGACCTACTGTCCTA |
| 35 | IL-11 Rev | GCCCAGTCAAGTGTCAGGTG |
| 36 | JUNB FW | ACTCATACACAGCTACGGGATACG |
| 37 | JUNB Rev | GGCTCGGTTTCAGGAGTTT |
| 38 | SERPINE1 FW | GAGACAGGCAGCTCGGATTC |
| 39 | SERPINE1 Rev | GGCCTCCCAAAGTGCATTAC |
| 40 | SMAD7 FW | ACCCGATGGATTTTCTCAAACC |
| 41 | SMAD7 Rev | GCCAGATAATTCGTTCCCCCT |
| 42 | Adapter P1 | /5Phos/AATTGTTCCCTACACGGACTGAATACTCTGGCCGTCGTTTTAC |
| 43 | Adapter P2 | /5Phos/GATCGTTCCCTACACGGACTGAATACTCTGGCCGTCGTTTTAC |
| 44 | Adapter P3 | /5Phos/CATGCTTCCCTACACGGACTGAATACTCTGGCCGTCGTTTTAC |
| 45 | Adapter P4 | /5Phos/AGCTGTTCCCTACACGGACTGAATACTCTGGCCGTCGTTTTAC |
| 46 | Adapter P5 | /5Phos/CTTCCCTACACGGACTGAATACTCTGGCCGTCGTTTTAC |
| 47 | Adapter P6 | CAGGAAACAGCTATGACAGTATTCAGTCCGTGTAGGGAAC |
| 48 | Fwd primer P1 | GTAAAACGACGGCCAG |
| 49 | Rev primer P1 | CAGGAAACAGCTATGAC |

using the 2× qPCR SyGreen Mix (PCR Biosystems) on a BioRad CFX96 real-time PCR detection system, following the manufacturer's guidelines. Relative gene expression levels were calculated using the ΔΔCt method, normalized to *GAPDH* expression, and quantified relative to the control condition. The sequence of the oligonucleotides used in this experiment is presented in Table 2, oligo no 30–41.

## STEEL-seq using PacBio sequencing
To test the functionality of the STEEL method with PacBio sequencing, extracted and purified DNA from TK6 lymphoblast cells was treated with Sloppymerase and the DNA was sequenced. The DNA was extracted with the DNeasy Blood & Tissue Kit (Qiagen) according to the manufacturer's instructions. Before treatment 8,7 μg of purified DNA from untreated wild type TK-6 cells was incubated with 25 units of the nickase Nt.BsmAI for 45 min at 37 °C and then heat-inactivated for 20 min at 65 °C to introduce nicks. The reaction was prepared in 1,5 ml eppendorf tubes with 1× NEBuffer 2.1 (New England Biolabs), a mixture of dNTPs (0.1 mM dCTP, 0.2 mM dGTP, 0.05 mM dTTP) and 0.05 mM Biotin-11-dUTP, 0.1 mM MnCl$_2$, and 0.035 μg/μl Sloppymerase. The DNA was added to the reaction at a concentration of 130 ng/μl. The sample was then incubated at 37 °C for 90 min followed by heat inactivation for 20 min at 75 °C. To fill any potential gaps, the sample was treated with 10 U Klenow fragment (3'–5' exo-) (New England Biolabs) after adding 0.2 mM dATP, 0.1 mM dCTP, and 0.15 mM dTTP to account for different concentrations of dNTPs. The sample was incubated for 30 min at 37 °C, followed by heat inactivation for 20 min at 75 °C

The sample was digested with a combination of EcoRI-HF, BamHI-HF, NcoI-HF, and HindIII-HF (New England Biolabs) (10 U/enzyme) for 60 min at 37 °C followed by a buffer exchange to ligation buffer with Amicon Ultra-0.5 Centrifugal Filter Unit according to the manufacturer's protocol. In the next step, a mixture of adapters (P1, P2, P3, P4, P5, and P6) with unique overhangs matching the different restriction sides and 30 U of T4 DNA ligase was added. The sample was then incubated overnight end over end at 4 °C. To extract the manipulated DNA fragments containing biotinylated dUTPs, a Dynabeads™ kilobaseBINDER™ Kit (Thermo Fisher Scientific) was used with a 3× higher concentration of beads than the manufacturer's recommendation to ensure a high yield, otherwise the beads were prepared and washed according to the original protocol. The solution with biotinylated DNA was incubated with the beads end-over-end overnight at 4 °C and an additional 2 h at RT the next morning. The bead solution was washed according to the manufacturer's protocol before they were eluted in ddH$_2$O at 75 °C for 5 min. The concentration of the recovered DNA was measured with nanodrop for PCR amplifciation. The PCR reaction was mixed according to the manufacturer's protocol with 175 ng of DNA, 0.2 mM dNTPs, 0.5 μM forward primer P1 and reverse primer P2, 1,5 mM MgCl$_2$, and 1,25 U Taq polymerase. The DNA was amplified in a thermocycler for 20 cycles at 94 °C for 45 s, at 51 °C for 30 s, and 2.5 min at 72 °C. The samples were run on a Novex™ TBE Gels, 10% (Thermo Fisher Scientific) for quality control followed by PCR cleanup with MinElute PCR Purification Kit (Qiagen) according to the manufacturer's instructions and eluted in nuclease free water. After purification, the samples were run on an agarose gel and the concentration was measured with nanodrop before they were sent to PacBio sequencing. The sequencing was performed by SNP & Seq technology platform using PacBio Revio system to generate 1 million HiFi reads. The sequence of the oligonucleotides used in this experiment is presented in Table 2, oligo no 42–49.

## STEEL-seq using Nanopore sequencing
DNA extracted from HaCaT keratinocyte cells was either left untreated, or nicked with 10 U of Nt.BsmAI for 60 min at 37 °C and thereafter heat-inactivated for 20 min at 65 °C. The DNA was added to 1× NEBuffer 2.1, a mixture of dNTPs (0.2 mM dCTP, 0.2 mM dGTP, 0.15 mM dTTP)

(Thermo Fisher Scientific) and 0.05 mM Biotin-11-dUTP (Jena Bioscience), 0.1 mM MnCl$_2$ (Merck) and 0.035 μg/μl Sloppymerase for 60 min at 37 °C and heat-inactivated for 20 min at 75 °C. To fill any remaining gaps, the samples were then treated with 10 U Klenow Fragment (3' → 5' exo-), 60 U T4 DNA ligase and 0.1 mM dATP for 60 min at 37 °C, followed by heat inactivation for 20 min at 75 °C. The DNA was fragmentized with 20 U of PmeI and 40 U of PmlI (New England Biolabs) for 60 min at 37 °C with 1 volume of 1× cutsmart buffer and heat-inactivated for 20 min at 65 °C. The samples were then sequenced with Oxford Nanopore. The sequencing was performed by SNP&SEQ technology platform using a PromethION system and two flow cells.

## Data analysis
**Alignment of STEEL-seq reads.** GRCh38 was used as reference sequence. Illumina reads were aligned with strobealign[45] using default settings. Nanopore and PacBio HiFi reads were aligned with minimap2[46] using -ax map-ont and -ax map-hifi, respectively.

**Break detection.** We developed a Python script that finds potential SSB sites in aligned reads from STEEL-seq libraries prepared without dATP in the reaction mix. The main signature of Sloppymerase activity in this setting is substitution of adenine with a different nucleotide– typically guanine, but we do not impose this as a restriction. The script thus starts by searching each aligned read for regions in which all adenines appear mutated (substituted or deleted). As also the reverse strand may have been sequenced, it also searches for regions where all thymines appear mutated. We call these regions *events* below. Supplementary alignments are ignored in the current version of the script. Used software libraries include pysam (https://github.com/pysam-developers/pysam) and pyfaidx (https://doi.org/10.7287/peerj.preprints.970v1).

**Filtering.** Since sequencing errors can easily conspire to look like Sloppymerase activity, we apply the following filters to ensure we remove most false positives.

1) Aligned reads with an overall mutation rate (counting substitutions, insertions, and deletions) above 10% are removed. Since both sequencing errors and mutations caused by Sloppymerase are counted, this threshold is set to a value well above the typical sequencing error rate.

2) An event is only kept if the number of substituted or deleted adenines (or thymines when on the reverse strand) is at least five for PacBio HiFi and Nanopore reads, three for Illumina reads from TK6 and four for Illumina reads from HaCaT.

3) The average base quality of substituted bases must be at least 10.

4) For Nanopore reads, at least three of the mutations in an event must be substitutions (i.e., not deletions).

Choice of which filters to implement and the values for filtering thresholds were informed by comparing the script output to a list of manually annotated ground truth events on a 1 Mbp region of a nickase-treated sample.

**Output.** The script computes the regions within which the SSB site must be located. These are between the last unmutated adenine upstream of the mutated region and its first mutated nucleotide for the case of mutated adenines, and between the last mutated nucleotide and the first unmutated thymine downstream of the region for the case of mutated thymines. The output is a BED track with extra annotations in a format that IGV[47] understands and displays (including read name, mapping quality, base qualities). In a postprocessing step, the BED file is sorted using BEDTools[48] and duplicate entries are removed. Additional output is a BAM file with the subset of the input reads on which events were detected.

**Single strand breaks annotations.** SSB hits from the Illumina sequencing were deduplicated based on the position. PacBio and NanoPore data were not deduplicated as they correspond to unique break events. Non-canonical chromosomes were filtered out before annotation. For Illumina data, non-irradiated and non-treated cells data were used as references and, Sloppymerase-treated samples were filtered using the references. The ChIPseeker, a R Bioconductor package, was used to annotate genomic regions (promoter, 5′UTR, 3′ UTR, exon, intergenic regions) and visualize forward and reverse strand breaks[49]. Notebooks for data exploration, preprocessing, and annotation are available on GitHub (https://github.com/barslmn/sloppymerase-annotations).

**Differentially expressed genes (DEGs) analysis.** RNA-seq data analysis was performed by nf-core/rnaseq pipeline, v3.18.0[50]. Briefly, after adapter removal and trimming (Trim Galore v.0.6.10) of low-quality reads, reads alignment to the GRCh38 reference genome was performed by using the Spliced Transcripts Alignment to a Reference (v2.7.11b) aligner. Transcript-level quantification was conducted with Salmon (v.1.10.3). Differential gene expression levels were identified by DESeq2 package (v3.20) in R (v4.4.2). Genes with a log2 fold change >1.5 were identified as DEGs. Non-expressed genes were defined as having 0 gene count in all four conditions.

**Statistics and reproducibility.** PAGE analysis of Sloppymerase-treated hairpins, with 3 and 4 nucleotides, has been performed routinely by two different individuals, yielding similar results. We use this method to validate the activity of Sloppymerase for each batch produced, and before each STEEL-seq experiment.

### Reporting summary
Further information on research design is available in the Nature Portfolio Reporting Summary linked to this article.

## Data availability
The raw sequencing reads used in this study are available from the European Nucleotide Archive (ENA) using accession PRJEB79373. Source data are provided with this paper.

## Code availability
The break-detection software and a pipeline for replicating results from this paper are available on GitHub (https://github.com/NBISweden/Sloppymerase/)[51]. Notebooks for data exploration, preprocessing, and annotation are available on GitHub. (https://github.com/barslmn/sloppymerase-annotations). The code for break annotation and plotting is available on Zenodo (https://doi.org/10.5281/zenodo.15730032)[52]. Used software libraries include pysam (https://github.com/pysam-developers/pysam) and pyfaidx (https://doi.org/10.7287/peerj.preprints.970v1).

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

## Acknowledgements

This study has been funded by grants from the Swedish Cancer Foundation (22 2306 Pj, O.Sö.). Illumina Sequencing was performed by the SNP&SEQ Technology Platform in Uppsala and Stockholm. The facility is part of the National Genomics Infrastructure (NGI) Sweden and Science for Life Laboratory. The NGI Platform is funded by Science for Life Laboratory, the Swedish Research Council and the Knut and Alice Wallenberg Foundation. For PacBio and Nanopore sequencing the authors would like to acknowledge support of the National Genomics Infrastructure (NGI)/Uppsala Genome Center, NAISS, and UPPMAX for providing assistance in massive parallel sequencing and computational infrastructure. Sequence analysis was supported by the SciLifeLab & Wallenberg Data Driven Life Science Program, Knut and Alice Wallenberg Foundation (grants: KAW 2020.0239 and KAW 2017.0003), and by the National Bioinformatics Infrastructure Sweden (NBIS) at SciLifeLab. Proteomic analysis was supported by the Spatial Mass Spectrometry unit at SciLifeLab.

## Author contributions

L.W., J.H., A.S., F.A.S., E.T.J., A.D., and B.S. performed experiments, the vast majority by L.W. M.M., Y.E., B.S., and W.S. performed sequence analysis. Y.E., X.C., J.A.E., M.L., J.L., O.Sp., and O.Sö. supervised experiments and/or sequence analysis. LW & OSö drafted the first version of the manuscript; all coauthors contributed to the final version of the manuscript. O.Sö. conceived the project, designed Sloppymerase and coordinated all the work in the project.

## Funding

## Competing interests

L.W., J.H., and O.Sö. are inventors on the patent application (WO Patent 2022/093091 A1) covering the design and use of Sloppymerase. The remaining authors declare no competing interests.
