## [Transparent Peer Review file · Nature Communications]

Precise mapping of single-stranded DNA breaks by sequence-templated erroneous DNA polymerase end-labelling

Corresponding Author: Professor Ola Söderberg

Version 0:

Reviewer comments:

Reviewer #1

(Remarks to the Author)

The manuscript by Wenson et al. aims to develop a novel method for detecting single-stranded DNA breaks (SSBs), which are common DNA lesions. Current methods for detecting SSBs are imprecise and can cause secondary damage. To address this issue, the researchers engineered a chimeric DNA polymerase called "Sloppymerase," which consists of DNA polymerases derived from *Saccharomyces cerevisiae* and *Escherichia coli*, capable of replicating DNA in the absence of specific nucleotides. This polymerase introduces mismatches at positions downstream of SSBs, enabling accurate mapping of these breaks through a sequencing-based approach called STEEL-seq. The method is compatible with multiple sequencing platforms, ensuring the precise identification of SSBs without causing secondary damage. The authors emphasize the potential of Sloppymerase to provide robust and precise detection of DNA damage, which is crucial for understanding genomic integrity and identifying damage caused by external or internal factors.

Amplified DNA strands generated by Sloppymerase can be analyzed using various sequencing methods, demonstrating the versatility of this approach. These findings could expand our understanding of the distribution of SSB sites and enhance our knowledge of cellular repair mechanisms for such lesions.

Specific comments:

1. The engineering of the error-prone polymerase is a key innovation in this study; however, the diagram explaining the characteristics of the chimeric enzyme is insufficient. Although the main text explains this well, Fig. 1 should be redrawn with more detailed features.
2. The predicted structure of Sloppymerase is shown in Fig. 2C. Have the structures of the individual polymerases been reported? I suggest including the structures of each polymerase along with the predicted structure of the chimeric enzyme. Additionally, a diagram of the domain structure would improve clarity. A protein band profile, such as Coomassie staining of purified protein, should also be included in the main or supplementary figures.
3. HaCaT cells were used coupled with radiation method to obtain DNA containing SSBs. The rationale for using HaCaT cells should be explained in the main text.
4. Regarding the sensitivity of the proposed method, what is the minimum amount of genomic material required? Since identifying heterogeneity in SSBs among populations, such as cancer cells, is of interest, it is crucial to provide the sensitivity of this method. Can it be applied to single-cell analysis?
5. The authors successfully mapped SSB locations and compared results between irradiated and non-irradiated TK6 cells. Are there any previous studies that support these findings? Why are irradiation-induced SSBs less frequently located in promoter regions?
6. Although current methods are inadequate for distinguishing between SSBs and double-stranded breaks (DSBs), I strongly recommend the authors compare their results with those obtained using existing methods. Given the low frequency of DSBs,

such a comparison would provide valuable insights.

(Remarks on code availability)

Reviewer #2

(Remarks to the Author)

Wenson et al. developed a novel detection method for single-stranded DNA breaks (SSBs) using a chimeric DNA polymerase called Sloppymerase. They aimed to engineer a highly error-prone DNA polymerase with high tolerance to modified nucleotides, lacking 3'-5' exonuclease activity but retaining 5'-3' exonuclease activity. They combined DNA polymerase η , a translesion DNA polymerase, with the 5'-3' exonuclease domain of E. coli DNA polymerase I. They claimed that Sloppymerase could be applicable for sequence-templated erroneous end-labeling sequencing (STEEL-seq) of SSBs. However, there is a gap in the logical flow of the manuscript. My major and minor concerns are as follows (not in chronological order):

1. Figure 1 shows a gap in the top strand, but SSB typically refers to a nick, both in the manuscript and figure legend. Please clarify this detail. Can Sloppymerase erroneously introduce dNTPs at a DNA gap?
2. Do the catalytic mutants (dead forms) of Polymerase I and η function in Sloppymerase?
3. Lines 182-183: The authors demonstrated protein expression using LC-MS, but direct methods such as SDS-PAGE and western blot are necessary to confirm protein expression and purity.
4. Figure 2 and Supplementary Figure 3 need quantification of dNTP incorporation, especially since Sloppymerase is a newly engineered protein.
5. Additionally, I am curious about the processivity of Sloppymerase. For example, to evaluate Sloppymerase's enzymatic activity, the authors might test consecutive nucleotide bases (e.g., AAAA) or GC-rich sequences downstream of the SSB sites.
6. Figure 2C: Superimposed images would help illustrate the differences between WT polymerases.
7. Figure 2D: Using Sloppymerase, the authors confirmed their findings from a previous study that DNA polymerase can label DNA damage in fixed/permeabilized cells. I believe this figure does not suggest novel characteristics of Sloppymerase because there is no comparison or indication of the advantages of this new enzyme. It seems like preliminary data.
8. Lines 223-229 & Fig. 3: A schematic figure could improve readability.
9. In Figure 3A, top, sequence 1 shows no mutation.
10. Are there specific patterns when dNTPs are incorporated? A sequence logo might improve clarity. Additionally, how many indels are introduced? Were the indels ignored in further analysis?
11. Lines 244-248: Are there any figures or tables showing data such as unique reads, total read numbers, or random dNTP incorporation?
12. Lines 248-250: I think this is a bold statement without experimental evidence.
13. A pairwise (or relevant) comparison between STEEL-seq and one or two existing methods is necessary, as well as a comparison with an error-prone DNA polymerase.
14. Lines 269-272: The authors set a threshold for five or three consecutive dA replacements. How was this threshold determined? Also, in line 783, the authors mention a Q score >10. A systematic or quantitative assessment of the filtering process would be beneficial.
15. In Figure 5, the authors analyzed SSB hotspots, but the statistical details (e.g., coverage per region) seem insufficient to support these claims.
16. Since the authors used Nanopore and PacBio, were any allelic coincidences observed for SSB, or were they purely stochastic events?

(Remarks on code availability)

Reviewer #3

(Remarks to the Author)

In this manuscript, the authors describe the creation and characterization of a new chimeric DNA polymerase, which they name Sloppymerase. This enzyme consists of the budding yeast translesion polymerase (Pol) η and the 5'-3' exonuclease domain of the E. coli Pol I. Pol η , on its own, demonstrates low processivity, being able to polymerize only a few nucleotides before stopping; addition of a domain that bears 5'-3' exonuclease activity to it is expected to increase its processivity. The authors confirm that Sloppymerase indeed has both polymerase and exonuclease activity. They further demonstrate that the enzyme is capable of efficiently polymerizing DNA in the absence of dATP as well as incorporating biotinylated nucleotides. Next, the authors investigate the mutational profile that Sloppymerase generates when used in the absence of specific dNTPs, showing that purines are generally replaced by purines and pyrimidines by pyrimidines. Insertions and deletion are also often observed. Finally, the authors develop a technique to map single-strand breaks (SSBs) genome-wide that relies on the mutational signature of Sloppymerase, which they call STEEL-Seq (Sequence-templated erroneous end-labelling sequencing). As proof of principle, they digest human genomic DNA with the nicking endonuclease Nt.BsmAI, perform nick translation with Sloppymerase in the absence of dATP and sequence the resulting DNA by different sequencing technologies. A SSB is identified as a read that contains three or more replaced dATP, depending on the sequencing platform used to sequence the library. The position of the SSBs should lie somewhere between the last remaining dATP and the first replaced dATP. As expected, the authors observe significantly more predicted

SSB-containing reads in the samples treated with Nt.BsmAI than those that were not and that the majority can be mapped to the expected sites cut by the nicking enzyme. Finally, the authors use STEEL-Seq to investigate the properties of endogenous SSBs and report a potential accumulation of this type of lesions to promoter-proximal regions.

I find the development and use of a “sloppy” polymerase with a distinctive mutational signature a refreshing and ingenious approach to mapping SSBs, which circumvents the obligatory shortcomings of other technologies that do the same by capturing 3'-OH groups. That said, I feel that the authors need to address several outstanding issues, which could even be done using the data that they already have, before I can recommend publication of the manuscript in Nature Communications.

- The authors seem so assign the properties of Sloppymerase, with regards to how it polymerizes DNA, to this enzyme specifically. Yet, most likely, the same would apply to Pol eta, if this enzyme were used on a substrate like the nicked hairpin used in the manuscript. This point should be discussed somewhere.
- For the purpose of making STEEL-seq libraries, and other prospective applications, it would be interesting to estimate, properly, the speed of Sloppymerase.
- In Figure 2B, why is a positive control with all 4 dNTPs not included?
- In Figure 3B, the authors should explain why most of the reads do not show a substitution of an A to a G. For example, the first A is converted to a G only in 44.8% of the reads. Is this observed because the nicked hairpin is not extended at all by Sloppymerase in X% of cases and therefore the “original” synthetic substrate/oligo is sequenced? If this is the case, the authors should estimate the efficiency with which Sloppymerase can extend a 3'-OH in the context of a nick or gap.
- Further characterization of the high throughput data should be done. The authors should extract additional interesting benchmarking metrics from the data that they already have, such as: 1) how many Nt.BsmAI sites can, and cannot, be detected?, 2) are there specific properties associated with the sites that can be detected more/less efficiently?, number of reads per site (normalized by sequencing depth) etc. This would further validate the strengths, and weaknesses, of their approach.
- No experiments are shown that address how sensitive STEEL-Seq is. This could be accomplished by titrating the amount of Nt.BsmAI cuts in the DNA used to make STEEL-Seq libraries.
- Optional: From the samples that were digested with Nt.BsmAI, could the authors not quantify the actual number of endogenous SSBs per genome using a strategy similar to <https://www.nature.com/articles/s41467-019-10332-8>? And compare these numbers to those reported by <https://www.nature.com/articles/362709a0>.
- The authors should at least discuss why the number of reads identified as SSB is a very small percentage of total number reads detected per sample.
- Fig 3B and 4 are unnecessarily overly complicated. The authors should simplify these figures so to express more clearly their message.
- Fig. 5. I do not think that showing the distribution of SSBs at the different functional elements of/around a gene as a pie chart is a good strategy, because, as far as I can tell, this does not consider the length of such regions. Therefore, I would suggest the authors rework this figure rather to show the # SSBs per kb in a bar chart.
- There are a couple of typos/text errors: 1) line 333, “that the a” and 2) legend of Table 1, “excluded”.

(Remarks on code availability)

The code is well documented on GitHub. I could clone the repository and easily create the necessary environment with all the requirements using the supplied environment.yml file. I did not try to run their custom code with the supplied raw data.

Version 1:

Reviewer comments:

Reviewer #1

(Remarks to the Author)

The authors have satisfactorily addressed all suggested experiments and textual changes. I therefore recommend this manuscript for publication.

(Remarks on code availability)

Reviewer #2

(Remarks to the Author)

This is a point-by-point response to a previously submitted manuscript. The authors have adequately responded to all of my previous comments. I find the revised manuscript significantly improved and I think it is now acceptable for publication.

Very minor point:

In Figure 3a, 2nd row, the red-colored "T" in the sequence 'GGATCCGCCAAGCTTCGAGCTGAATTCTGCAGTACATTAATTGGGT"**T**"TGGG' remains unmutated.

(Remarks on code availability)

Reviewer #3

(Remarks to the Author)

The authors have constructively addressed my original comments and I am happy with the changes that they have brought to the manuscript.

The have just three small remarks:

* The authors did not really address my comment about the sensitivity of STEEL-seq "No experiments are shown that address how sensitive STEEL-Seq is. This could be accomplished by titrating the amount of Nt.BsmAI cuts in the DNA used to make STEEL-Seq libraries." This comment was more about how well STEEL-Seq would detect SSBs, let's say, at a specific location in the context of genomic DNA rather than how efficiently their hairpin substrate is extended by Sloppymerase. In other words, if they mixed undigested and Nt.BsmAI-treated genomic DNA at different ratios, when would the Nt.BsmAI sites not be detected anymore? I leave it to the authors to decide whether they think this information would be useful for the community (I think it would).

* I would recommend to the authors to run their manuscript through a spell-/grammar-/style- check tool because there are still some typos and styling issues here and there.

* The link <https://www.nature.com/articles/362709a0> does work, I think Word included the final full-stop of the paragraph in it previously, and that is why it did not work.

(Remarks on code availability)

The code is well documented on GitHub and enough comments are included. I could clone the repository and create the necessary environment with all the requirements using the supplied environment.yml file. I could not run the pipeline because I could not find the raw data at <https://www.ebi.ac.uk/ena> using accession PRJEB79373.

REVIEWER COMMENTS

Reviewer #1 (Remarks to the Author):

The manuscript by Wenson et al. aims to develop a novel method for detecting single-stranded DNA breaks (SSBs), which are common DNA lesions. Current methods for detecting SSBs are imprecise and can cause secondary damage. To address this issue, the researchers engineered a chimeric DNA polymerase called "Sloppymerase," which consists of DNA polymerases derived from *Saccharomyces cerevisiae* and *Escherichia coli*, capable of replicating DNA in the absence of specific nucleotides. This polymerase introduces mismatches at positions downstream of SSBs, enabling accurate mapping of these breaks through a sequencing-based approach called STEEL-seq. The method is compatible with multiple sequencing platforms, ensuring the precise identification of SSBs without causing secondary damage. The authors emphasize the potential of Sloppymerase to provide robust and precise detection of DNA damage, which is crucial for understanding genomic integrity and identifying damage caused by external or internal factors.

Amplified DNA strands generated by Sloppymerase can be analyzed using various sequencing methods, demonstrating the versatility of this approach. These findings could expand our understanding of the distribution of SSB sites and enhance our knowledge of cellular repair mechanisms for such lesions.

Specific comments:

1. The engineering of the error-prone polymerase is a key innovation in this study; however, the diagram explaining the characteristics of the chimeric enzyme is insufficient. Although the main text explains this well, Fig. 1 should be redrawn with more detailed features.

We have redrawn the figure, added more details, and think this have made it more clear.

2. The predicted structure of Sloppymerase is shown in Fig. 2C. Have the structures of the individual polymerases been reported? I suggest including the structures of each polymerase along with the predicted structure of the chimeric enzyme. Additionally, a diagram of the domain structure would improve clarity. A protein band profile, such as Coomassie staining of purified protein, should also be included in the main or supplementary figures.

We agree and have added structures of the individual polymerases into the figure. We have included Coomassie and WB staining and a diagram of the domain structure of Sloppymerase, as **supplementary figure 2**.

3. HaCaT cells were used coupled with radiation method to obtain DNA containing SSBs. The rationale for using HaCaT cells should be explained in the main text.

We use HaCaT cells often in the lab, as it is a good cell for analysis of signal transduction. We used two different cell lines in this project, HaCaT and TK-6 cells. We have in the experiments

performed in the revision used HaCaT cells to monitor the effect of TGF β treatment, as this cell line respond well to TGF β treatment.

4. Regarding the sensitivity of the proposed method, what is the minimum amount of genomic material required? Since identifying heterogeneity in SSBs among populations, such as cancer cells, is of interest, it is crucial to provide the sensitivity of this method. Can it be applied to single-cell analysis?

We agree that it would be of great value to be able to apply the method to single-cell analysis. Sloppymerase-treatment should be able to be down scaled to very low amounts of DNA, although for this paper we had to focus on a few applications. But we have confirmed that it is possible to perform the Sloppymerase-treatment in fixed permeabilized cells (**supplementary figure 5**), which would provide the ability to do the labeling prior to DNA extraction. Hence, we think that the protocol could readily be adopted for single-cell analysis. We have added a sentence in the discussion section about this.

5. The authors successfully mapped SSB locations and compared results between irradiated and non-irradiated TK6 cells. Are there any previous studies that support these findings? Why are irradiation-induced SSBs less frequently located in promoter regions?

The amount of naturally occurring SSBs are more frequent in promoter regions, on top of these comes the SSBs generated by the irradiation. It seems that they are more randomly distributed. Hence, combined the frequency of SSBs will be lower. However, as irradiation mainly causes DSBs the amount of SSBs is not very high. We agree with the reviewer that there might be better models to monitor changes in SSB frequency than irradiation and have hence performed a study to evaluate if frequency of SSBs in promoter regions are connected with gene expression. For this experiment we have treated HaCaT cells with TGF β and monitor early response genes, by transcriptome analysis (RT-PCR and sequencing) to correlate gene expression with SSBs at the promoter regions. In addition, we also pretreated the cells with a z-DNA inducing drug, as we hypothesized that formation of z-DNA may be a consequence of inhibiting SSBs at the promoter regions. We have added a new figure and text, both in results and discussion sections, presenting the new data.

6. Although current methods are inadequate for distinguishing between SSBs and double-stranded breaks (DSBs), I strongly recommend the authors compare their results with those obtained using existing methods. Given the low frequency of DSBs, such a comparison would provide valuable insights.

We agree and have added a comparison with data from another method SSINGLE, describing similarities and the advantages and disadvantages compares to STEEL-seq, discussion section.

Reviewer #2 (Remarks to the Author):

Wenson et al. developed a novel detection method for single-stranded DNA breaks (SSBs) using a chimeric DNA polymerase called Sloppymerase. They aimed to engineer a highly

error-prone DNA polymerase with high tolerance to modified nucleotides, lacking 3'-5' exonuclease activity but retaining 5'-3' exonuclease activity. They combined DNA polymerase η , a translesion DNA polymerase, with the 5'-3' exonuclease domain of E. coli DNA polymerase I. They claimed that Sloppymerase could be applicable for sequence-templated erroneous end-labeling sequencing (STEEL-seq) of SSBs. However, there is a gap in the logical flow of the manuscript. My major and minor concerns are as follows (not in chronological order):

1. Figure 1 shows a gap in the top strand, but SSB typically refers to a nick, both in the manuscript and figure legend. Please clarify this detail. Can Sloppymerase erroneously introduce dNTPs at a DNA gap?

We made break in the DNA strand a bit big, to make it visible. We have now remade the figure to make it clearer. We have tested Sloppymerase on both nicks and gaps, and see no difference in performance.

2. Do the catalytic mutants (dead forms) of Polymerase I and η function in Sloppymerase?

We assume that a catalytic dead form of Polymerase I, which removes proof reading, might possibly be used as an alternative to Sloppymerase. We did a few attempts in the beginning of this project didn't get it to work efficient enough. We then constructed the chimeric Sloppymerase that gave good results, and focused on this polymerase instead. We included a sentence in the discussion section that alternative error prone, catalytic dead polymerases might be a possible alternative to Sloppymerase in the STEEL-seq method

3. Lines 182-183: The authors demonstrated protein expression using LC-MS, but direct methods such as SDS-PAGE and western blot are necessary to confirm protein expression and purity.

We agree and have now included images of Comassi stained SDS-PAGE and Western blot as **supplementary figure 2**.

4. Figure 2 and Supplementary Figure 3 need quantification of dNTP incorporation, especially since Sloppymerase is a newly engineered protein.

We have now included a quantification of the **Figure 2**.

5. Additionally, I am curious about the processivity of Sloppymerase. For example, to evaluate Sloppymerase's enzymatic activity, the authors might test consecutive nucleotide bases (e.g., AAAA) or GC-rich sequences downstream of the SSB sites.

I have to admit that I shared this curiosity, a good suggestion that I think might be helpful for the readers interested in polymerases. We designed a few different hairpins to test this and performed some additional sequencing (**Figure 3B**). Sloppymerase has no problem with GC-rich regions, but when we test 1, 2 or 4 consecutive A:s we observed an increase in deletions. For this experiment, we introduced the sequencing adapters a bit upstream, to provide an error rate for Sloppymerase when all dNPS are present, by comparing upstream and

downstream sequence, it is clear that the errors introduced are not a consequence of the PCR used for sequencing library prep. The error rate is a few percent, which is really astonishing.

6. Figure 2C: Superimposed images would help illustrate the differences between WT polymerases.

We agree and have added structures of the individual polymerases as **Figure 2C**.

7. Figure 2D: Using Sloppymerase, the authors confirmed their findings from a previous study that DNA polymerase can label DNA damage in fixed/permeabilized cells. I believe this figure does not suggest novel characteristics of Sloppymerase because there is no comparison or indication of the advantages of this new enzyme. It seems like preliminary data.

The purpose of this experiment was to confirm that Sloppymerase could label DNA in fixed cells, which is expected, as it would allow for single cell analysis. This isn't the scope of the current paper, so we have moved the figure to **supplementary figure 5**.

8. Lines 223-229 & Fig. 3: A schematic figure could improve readability.

A schematic figure is presented as **supplementary figure 4**.

9. In Figure 3A, top, sequence 1 shows no mutation.

The top row is the sequence of the hairpin, toward which the reads are aligned to. Figure 3A is when all dNTPs are present, i.e. no substitutions expected.

10. Are there specific patterns when dNTPs are incorporated? A sequence logo might improve clarity. Additionally, how many indels are introduced? Were the indels ignored in further analysis?

We have replaced Fig 3B with the new sequencing data of the more complex hairpin you suggested (comment 5). We have included data on when all dNTPs are used and when dATP is omitted. We think this should make the image clearer, showing the performance of Sloppymerase. A is in most cases replaced by G, or deletions. For the analysis of Sloppymerase signature, in STEEL-seq, we look for missing or substituted A:s, so indels are accounted for.

11. Lines 244-248: Are there any figures or tables showing data such as unique reads, total read numbers, or random dNTP incorporation?

We included a detailed table of the data in the new fig 3B, and included information on total reads and how many were unique for all dNTPs and when dATP is omitted.

12. Lines 248-250: I think this is a bold statement without experimental evidence.

We agree and rephrase the sentence.

13. A pairwise (or relevant) comparison between STEEL-seq and one or two existing methods is necessary, as well as a comparison with an error-prone DNA polymerase.

The finding that SSBs are more prevalent in the promoter region is something also reported using another method for detection of SSBs, the SSiNGLe method. In order to compare our findings, and to provide new biological relevant data to the paper, we performed an experiment to evaluate if frequency of SSBs in promoter regions are connected with gene expression. For this experiment we have treated HaCaT cells with TGF β and monitor early response genes, by transcriptome analysis (qRT-PCR and sequencing) to correlate gene expression with SSBs at the promoter regions. In addition, we also pretreated the cells with a z-DNA inducing drug, as we hypothesized that formation of z-DNA may be a consequence of inhibiting SSBs at the promoter regions. We have added a new figure and text, and compare with published data from SSiNGLe, describing similarities and the advantages and disadvantages compares to STEEL-seq, in the discussion section.

14. Lines 269-272: The authors set a threshold for five or three consecutive dA replacements. How was this threshold determined? Also, in line 783, the authors mention a Q score >10. A systematic or quantitative assessment of the filtering process would be beneficial.

Since no fully annotated ground-truth dataset is available, a quantitative evaluation that would provide us with accurate estimates of sensitivity and specificity was not possible, and we focused on a qualitative assessment with the aim of reducing false positives. That is, to be able to show that our method works, we want to be sure that the events that we find actually represent SSBs. Not detecting events, on the other hand, is fine since we assume that all samples are affected equally, which means that the SSBs rates will be comparable. We can achieve our aim by using conservative filtering thresholds. To choose the thresholds, we manually annotated a 1 Mbp region of chromosome 1 (20-21 Mbp) in one of the nickase-treated samples. We only annotated events for which we were very confident that they represent Sloppymerase activity: They had to start just just down- or upstream of a nickase cut site, had to come from high-quality reads, and must have had no or very few mutations before and after the mutated region (otherwise it is more likely that all differences to the reference actually are sequencing errors). We then chose the thresholds as low as necessary for all of these high-confidence events to be found, but not higher than that. We avoided annotating anything else (positives or negatives): For example, mutations that looked like Sloppymerase activity but were not in the vicinity of a nickase motif might be caused by sequencing errors, but could also have been caused by an SSB not induced by nickase. Also, an event involving very few mutated bases, but that is close to a nickase sites, can arise from a true SSB, but also from conspiring sequencing errors. The consequence is that the chosen thresholds are quite conservative and necessarily underestimate the true number of SSBs. However, this allows us to get a better signal for the purpose of comparing SSB rates of different samples to one another.

15. In Figure 5, the authors analyzed SSB hotspots, but the statistical details (e.g., coverage per region) seem insufficient to support these claims.

We have repeated the Illumina sequencing, STEEL-seq, to provide an additional set of experiment with many more reads (Figure 5C and Table 1). This experiment shows the same pattern of SSBs primarily being located to promoter region, as previous Illumina and Nanopore experiments Figure 5A and Table 1). We also went further, to determine if the frequency of SSB/nucleotide correlate to activity state of promoter. The data shows that the frequency is higher in active promoters. The finding is in agreement with published data using the SSiNGLe method. We added a section in discussion about this.

16. Since the authors used Nanopore and PacBio, were any allelic coincidences observed for SSB, or were they purely stochastic events?

It is an interesting idea to investigate if the SSBs have a different frequency at different parts of the genome. To test this, we analyzed if the frequency differs in genes that are not expressed and observed a correlation with gene expression. The data is now presented in **Figure 5C**.

Reviewer #3 (Remarks to the Author):

In this manuscript, the authors describe the creation and characterization of a new chimeric DNA polymerase, which they name Sloppymerase. This enzyme consists of the budding yeast translesion polymerase (Pol) eta and the 5'-3' exonuclease domain of the E. coli Pol I. Pol eta, on its own, demonstrates low processivity, being able to polymerize only a few nucleotides before stopping; addition of a domain that bears 5'-3' exonuclease activity to it is expected to increase its processivity. The authors confirm that Sloppymerase indeed has both polymerase and exonuclease activity. They further demonstrate that the enzyme is capable of efficiently polymerizing DNA in the absence of dATP as well as incorporating biotinylated nucleotides. Next, the authors investigate the mutational profile that Sloppymerase generates when used in the absence of specific dNTPs, showing that purines are generally replaced by purines and pyrimidines by pyrimidines. Insertions and deletion are also often observed. Finally, the authors develop a technique to map single-strand breaks (SSBs) genome-wide that relies on the mutational signature of Sloppymerase, which they call STEEL-Seq (Sequence-templated erroneous end-labelling sequencing). As proof of principle, they digest human genomic DNA with the nicking endonuclease Nt.BsmAI, perform nick translation with Sloppymerase in the absence of dATP and sequence the resulting DNA by different sequencing technologies. A SSB is identified as a read that contains three or more replaced dATP, depending on the sequencing platform used to sequence the library. The position of the SSBs should lie somewhere between the last remaining dATP and the first replaced dATP. As expected, the authors observe significantly more predicted SSB-containing reads in the samples treated with Nt.BsmAI than those that were not and that the majority can be mapped to the expected sites cut by the nicking enzyme. Finally, the authors use STEEL-Seq to investigate the properties of endogenous SSBs and report a potential accumulation of this type of lesions to promoter-proximal regions.

I find the development and use of a “sloppy” polymerase with a distinctive mutational signature a refreshing and ingenious approach to mapping SSBs, which circumvents the obligatory shortcomings of other technologies that do the same by capturing 3'-OH groups. That said, I feel that the authors need to address several outstanding issues, which could even be done using the data that they already have, before I can recommend publication of the manuscript in Nature Communications.

• The authors seem so assign the properties of Sloppymerase, with regards to how it polymerizes DNA, to this enzyme specifically. Yet, most likely, the same would apply to Pol eta, if this enzyme were used on a substrate like the nicked hairpin used in the manuscript. This point should be discussed somewhere.

Yes, we expect that the polymerizing property of Sloppymerase will be similar as Pol eta, the difference is that Sloppymerase contains 5'-3' exonuclease that allows it to degrade the strand in front. This allows it to polymerize a longer stretch. We added a sentence in the discussion section that also other error prone DNA polymerases, with 5'-3' exonuclease activity might be used for STEEL-seq

• For the purpose of making STEEL-seq libraries, and other prospective applications, it would be interesting to estimate, properly, the speed of Sloppymerase.

We have included a quantification of the gel images, Figure 2, to give an estimate of speed.

• In Figure 2B, why is a positive control with all 4 dNTPs not included?

In Fig 2A we saw that -dCTP was able to extend the hairpin, almost as good as with all dNTP. For Fig 2B, we didn't compare towards all dNTP but only determined if Sloppymerase could extend the hairpin when other nucleotides were missing. As we saw that -dATP gave the strongest band, and because the Sanger sequencing gave less deletions compared to -dCTP. We choose -dATP for the STEEL-seq approach. To better see the performance of Sloppymerase, we redesigned the hairpin, so it contains A, AA and AAAA, as well as a stretch of GC to determine if it affects the choice of inserted nucleotide. We designed the experiment, to get sequencing data also upstream the nick. We have now replaced the fig 2B with this new one, as it better shows how Sloppymerase function for both 3 and 4 dNTPs. This also allowed as to determine the error rate of Sloppymerase, when all nucleotides are present. The error rate (+/- STD) for Sloppymerase was found to be 4.6 +/- 3.4%. The error rate was highest for A: 9.0 +/- 3.3%, followed by T: 4.6 +/- 3.0%, G: 3.1 +/- 1.3% and C: 2.4 +/- 0.8%.

• In Figure 3B, the authors should explain why most of the reads do not show a substitution of an A to a G. For example, the first A is converted to a G only in 44.8% of the reads. Is this observed because the nicked hairpin is not extended at all by Sloppymerase in X% of cases and therefore the “original” synthetic substrate/oligo is sequenced? If this is the case, the authors should estimate the efficiency with which Sloppymerase can extend a 3'-OH in the context of a nick or gap.

Yes, the lower efficiency can be attributed to two things: that Sloppymerase didn't extend so far or that the template strand was amplified during the subsequent PCR step. Most likely, the second reason has the strongest impact, as the PAGE images show that most hairpins are extended. For the redesigned experiment, above, we designed the adapters to limit the amplification of the template DNA strand. The new figure shows that the vast majority of reads comes from the modified strand. We included a sentence clarifying this.

• **Further characterization of the high throughput data should be done. The authors should extract additional interesting benchmarking metrics from the data that they already have, such as: 1) how many Nt.BsmAI sites can, and cannot, be detected?, 2) are there specific properties associated with the sites that can be detected more/less efficiently?, number of reads per site (normalized by sequencing depth) etc. This would further validate the strengths, and weaknesses, of their approach.**

Your comment made us think of the best way to determine the actual frequency of SSB/nucleotide, and realized we had not used the full potential of our data. So, rather than enrich for Sloppymerase modified DNA fragments, which would give a better coverage of SSBs present but would not allow for direct measurement of SSB/nucleotide frequency, we choose not to enrich for modified DNA fragments. By counting SSBs and dividing by total length of sequenced nucleotides, we obtained a direct measurement of the frequency of SSBs. We believe the approach to directly determine the frequency of SSB has advantages over estimations based upon proportion of endogenous SSBs compared to Nt.BsmAI induced ones as this is based upon the assumption that all Nt.BsmAI are cut and that there are no other sites cut by the nickase.

• **No experiments are shown that address how sensitive STEEL-Seq is. This could be accomplished by titrating the amount of Nt.BsmAI cuts in the DNA used to make STEEL-Seq libraries.**

The quantification of the elongation of the oligos (**Fig 2**), shows that the efficiency is very high. Almost no original hairpins remain. The quantification shows an efficiency of around 75%, but it should be noted that this is based on the conversion of the original shorter hairpin to a fully extended. As Sloppymerase also generates deletions (the smear in between), the efficiency is most likely closer to 100%.

• **Optional: From the samples that were digested with Nt.BsmAI, could the authors not quantify the actual number of endogenous SSBs per genome using a strategy similar to <https://www.nature.com/articles/s41467-019-10332-8>? And compare these numbers to those reported by <https://www.nature.com/articles/362709a0>.**

Unfortunately, the link <https://www.nature.com/articles/362709a0> did not work. So we couldn't do the comparison. Instead, we compared to frequency reported for *E. coli* which is similar to what we observed.

- **The authors should at least discuss why the number of reads identified as SSB is a very small percentage of total number reads detected per sample.**

As we do not enrich for Sloppymerase modified DNA fragments, the low number reflects the frequency of SSBs in the genome. We have added some text in the discussion section to emphasize this.

- **Fig 3B and 4 are unnecessarily overly complicated. The authors should simplify these figures so to express more clearly their message.**

We agree that the images are a bit complicated, but feel it is important to show examples on how the reads look like. We changed the Fig 3B to show both Sloppymerase modifications with all dNTPs and without dATP. In addition, we also show a short stretch of unmodified sequence upstream, to highlight the action of Sloppymerase. We hope this modification improves clarity.

- **Fig. 5. I do not think that showing the distribution of SSBs at the different functional elements of/around a gene as a pie chart is a good strategy, because, as far as I can tell, this does not consider the length of such regions. Therefore, I would suggest the authors rework this figure rather to show the # SSBs per kb in a bar chart.**

This is a very good suggestion! In the original Figure 5, the first Pie chart represented the length of the region, but it is difficult for a reader to use the information. We made a bar chart, where the size in the x-axis correspond to size in the genome. On the y-axis we plotted the frequency of SSBs per sequenced nucleotide. We think this made the presentation so much better, thanks for the suggestion! We got intrigued by the enrichment of SSBs at promoter region and performed an experiment, with serum starved cells and stimulated with TGF β . We monitored early response genes and analyzed SSBs at these genes. As a reference we also analyzed SSBs at genes that were not expressed. We observed an enrichment at active promoters. The enrichment was present both upstream TSS and downstream, 5' UTR and 1st intron. We think this finding can shed light on the process of gene expression. We discuss this a bit in the discussion section.

- **There are a couple of typos/text errors: 1) line 333, "that the a" and 2) legend of Table 1, "excluded".**

Reviewer #3 (Remarks on code availability):

The code is well documented on GitHub. I could clone the repository and easily create the necessary environment with all the requirements using the supplied environment.yml file. I did not try to run their custom code with the supplied raw data.

Reviewer 2 found a typo in Figure 3:

Very minor point: In Figure 3a, 2nd row, the red-colored "T" in the sequence 'GGATCCGGCCAAGCTTCGAGCTGAATTCTGCAGTACATTAATTGGGT"**T**"TGGG' remains unmutated.

Response: We have now corrected the Figure.

Reviewer 3 raised two questions

* The authors did not really address my comment about the sensitivity of STEEL-seq "No experiments are shown that address how sensitive STEEL-Seq is. This could be accomplished by titrating the amount of Nt.BsmAI cuts in the DNA used to make STEEL-Seq libraries." This comment was more about how well STEEL-Seq would detect SSBs, let's say, at a specific location in the context of genomic DNA rather than how efficiently their hairpin substrate is extended by Sloppymerase. In other words, if they mixed undigested and Nt.BsmAI-treated genomic DNA at different ratios, when would the Nt.BsmAI sites not be detected anymore? I leave it to the authors to decide whether they think this information would be useful for the community (I think it would).

Response: We agree that it is important to know the efficiency of STEEL-seq. The experiments done with DNA-hairpins allowed us to determine the efficiency of Sloppymerase-modifications of SSBs, which is close to 100%. Determining the efficiency of detecting modified DNA sequences by titrating the amount of Nt.BsmAI cuts, the experimental setup suggested by the reviewer, will be dependent on the number of reads generated by the sequencing technology used. The more reads we obtain, the higher is the likelihood we will find Sloppymerase-modified Nt.BsmAI cuts, i.e., the sensitivity will be dependent on the number of reads. The cost for doing such experiment will be very high and provide limited value. We believe the experiment done with DNA-hairpins are cleaner and gives more information than the experiment suggested. Hence, we opt not to perform any additional experiments.

* I would recommend to the authors to run their manuscript through a spell-/grammar-/style-check tool because there are still some typos and styling issues here and there.

Response: We have done additional spell-/grammar-/style- check using Grammarly and with help from a native English spoken colleague.